# Multi-domain improves out-of-distribution and data-limited scenarios for medical image analysis

## Abstract

Current machine learning methods for medical image analysis primarily focus on developing models tailored for their specific tasks, utilizing data within their target domain. These specialized models tend to be data-hungry and often exhibit limitations in generalizing to out-of-distribution samples. In this work, we show that employing models that incorporate multiple domains instead of specialized ones significantly alleviates the limitations observed in specialized models. We refer to this approach as *multi-domain model* and compare its performance to that of specialized models. For this, we introduce the incorporation of diverse medical image domains, including different imaging modalities like X-ray, MRI, CT, and ultrasound images, as well as various viewpoints such as axial, coronal, and sagittal views. Our findings underscore the superior generalization capabilities of multi-domain models, particularly in scenarios characterized by limited data availability and out-of-distribution, frequently encountered in healthcare applications. The integration of diverse data allows multi-domain models to utilize information across domains, enhancing the overall outcomes substantially. To illustrate, for organ recognition, multi-domain model can enhance accuracy by up to 8% compared to conventional specialized models.

## 1 Introduction

In medical image analysis, existing machine learning approaches propose models to address wide range of problems (Gulshan et al., 2016; Irvin et al., 2019; Liu et al., 2020), which have been tailored for their designated applications and typically utilize data from a single target domain. This approach leads to data-intensive specialized models and show limited generalization capabilities. Proposed works in medical image analysis falls short of fully leveraging the diverse medical image data available. Various imaging modalities, such as X-rays, magnetic resonance imaging (MRI), computed tomography (CT), ultrasound (US), and positron emission tomography (PET), provide unique perspectives into different aspects of anatomy and pathology. X-rays excel in revealing bone structures and detecting fractures, while MRI scans provide detailed images of soft tissues like the brain, muscles, and organs. CT scans offer cross-sectional views, helping to identify internal injuries and complex conditions. US images are non-invasive and excel in real-time imaging, often used for monitoring pregnancies and examining internal organs, whereas PET provides metabolic information, aiding in cancer detection and localization. The combination of these imaging modalities is common in clinical practice and enhances diagnostic accuracy by providing complementary information that might not be evident in a single imaging method. Furthermore, in medical decision-making, clinicians often consider diverse viewpoints, since certain anomalies may be more apparent from one angle than another, ensuring a comprehensive understanding of the patient's condition and facilitating accurate diagnoses and effective treatment strategies.

Our work seeks to address a pivotal question: Can the integration of diverse image domains, such as medical imaging modalities or viewpoints, improve the generalization capabilities of models for a specific task in medical image analysis? To answer this question, the main contributions are as follows:

- We introduce a *multi-domain* model, with diverse medical image data domains, such as imaging modalities, like X-ray, MRI, CT, and US images, or various viewpoints, such as axial, coronal, and

sagittal views, This model uses the data from different imaging domains to train for a specific task with an off-the-shelf architecture. For comparison, we also train conventional specialized models for the same task using data exclusively from each individual domain.

- We evaluate the performance of specialized models in comparison to multi-domain model using three publicly available datasets, such as PolyMNIST Sutter et al. (2021), MedMNIST Yang et al. (2023) and ImageCLEFmedical Ionescu et al. (2022). We compare their accuracy for out-of-distribution (OOD) and data-limited scenarios, common in healthcare applications, showing that the integration of diverse data allows multi-domain models to enhance the overall outcomes in comparison to specialized models.

This work represents the first instance where multi-domain data is assessed for a single task, different from multi-task setups. It provides insights into the respective strengths and drawbacks of these models, showing the potential of diverse data domains in medical image analysis applications.

## 1.1 Related work

Recent years have witnessed a rise of foundation models, particularly in fields like natural language processing and computer vision. These models combine data from various domains and demonstrate exceptional generalization capabilities beyond their primary training tasks. In the sphere of LLMs, noteworthy examples include Tu et al. (2023); Zhang et al. (2023); Singhal et al. (2023); Yuan et al. (2021); Jin et al. (2019); Yuan et al. (2022); Lee et al. (2019); Rasmy et al. (2021); Luo et al. (2022); Li et al. (2020b); Yan et al. (2022), where some of these models utilized clinical notes and electronic health records in their development. Additionally, large vision models for healthcare have emerged, driving significant advancements across diverse applications (Qiu et al., 2023). For instance, Zhou et al. (2019); Azizi et al. (2021); Zhou et al. (2020); Huang et al. (2021); Sowrirajan et al. (2021); Zhang et al. (2022); Tiu et al. (2022); Nguyen et al. (2023) utilize self-supervised learning for different tasks using single imaging modality such as CT, MRI or X-rays. Efforts for multi-organ segmentation tasks also exist, such as Chen et al. (2021); Zhang et al. (2021); Xie et al. (2021b); Valanarasu et al. (2021); Hatamizadeh et al. (2022); Cao et al. (2022); Shi et al. (2023). These models have not only expanded in terms of their number of parameters and data handling capacities but have also consistently demonstrated remarkable performance once pre-trained. There have been some efforts to develop medical vision foundation models using diverse data, however, their widespread adoption remains limited.

When aiming to enhance generalization capabilities, an alternative approach to consider is multi-task learning (Caruana, 1997). Here, the goal is to improve the performance of a model while solving multiple related tasks simultaneously. The idea is that learning from multiple tasks can help the model capture shared patterns and representations, leading to better performance on each individual task. In contrast, in our work we focus on training with data from different domains without necessarily involving multiple tasks. As a practical example, our setup can be trained to identify abnormalities across CT, MRI, US and PET images. In contrast, multi-task learning focuses on detecting abnormalities while learning another related task within a single image domain, such as using MRI scans.

Several studies have explored the role of multi-modality approaches in healthcare contexts (Huang et al., 2021; Acosta et al., 2022; Tiu et al., 2022; Zhang et al., 2022; Yuan et al., 2023). However, these methods predominantly focus on integrating text with a single imaging modality, rather than incorporating data from various image domains. Closest work to ours is BenchMD (Wantlin et al., 2023), where they combined 19 publicly available datasets for 7 medical modalities, including 1D sensor data, 2D images, and 3D volumetric scans. In the case of 2D images, BenchMD combined data from diverse sources, including chest X-rays (CXR), mammograms, dermoscopic, and fundus images. They utilized widely-cited and large dataset as the primary source for each imaging modality, conducting evaluations of distribution shifts on a separate test set. Nevertheless, their analysis did not encompass an assessment of the models' generalization capability across various domains. In Table 1, we present a summary of our work, comparing it to related work based on their input, output, and the data needed for training and testing.

| | Input [# Images] | Output [# Tasks] | Dataset [# Domains] | Input instance [# Domains] |
|---|---|---|---|---|
| Specialized | Single | Single | Single | Single |
| Multi-task | Single | Multiple | Single | Single |
| Multi-modal | Multiple | Single | Multiple | Multiple |
| **Multi-domain** | **Single** | **Single** | **Multiple** | **Single** |

Table 1: Summary of our work, the multi-domain model, compared to related work of specialized, multi-task and multi-modal models based on their input, output, and the data needed for training and testing.

Our method is most related to works on multi-domain networks (Bilen & Vedaldi, 2017; Rebuffi et al., 2017; Rosenfeld & Tsotsos, 2018; Rebuffi et al., 2018), which focus on training a single network to handle image classification tasks across diverse domains. The primary objective is to develop a single network capable of compactly representing all domains with minimal task-specific parameters. These models introduce various architectures to incorporate diverse data domains into the model parameterization and assess their models' generalizability across different natural image datasets. In the healthcare domain, Mojab et al. (2020) proposed a self-supervised representation learning method that integrates multiple domains into the learned representations. However, our work differs from these methods by leveraging multi-domain data to train for a specific task using an off-the-shelf architecture.

## 2 Methods

The level of generalization that multi-domain models can achieve in scenarios involving out-of-distribution and limited data remains uncertain based on prior research involving large scale models applied to medical image analysis (Chen et al., 2019; Singhal et al., 2023; Wang et al., 2023; Zhao et al., 2023; Wantlin et al., 2023). To illustrate this idea, consider the following research question: Can a neural network trained on instances of a medical condition as observed through CT, PET, and X-ray images, provide accurate predictions when presented with MRI images, even in cases where this condition has been encountered infrequently in the training set from MRI images? In order to delve into the specifics, we aim to explore the potential for shared information across different data domains, such as imaging modalities or viewpoints. To achieve this, we will first introduce the datasets employed in this study and then outline the methods used to generate data diversity within these datasets, thereby allowing us to analyse the impact of diverse data domains on generalizability. For reproducability, we made our code publicly available in `https://anonymous.4open.science/r/multi_domain_medical-6686/`.

### 2.1 Datasets

Most existing datasets from various data domains are tailored to their specialized applications and these datasets lack commonalities that would allow for evaluating potential knowledge transfer. Thus, we present our results on the following datasets, which do not suffer from these challenges.

### 2.1.1 PolyMNIST

We start with the multi-modal benchmark PolyMNIST Sutter et al. (2021) to understand behaviours for different ablations. The PolyMNIST dataset consists of sets of ten MNIST digits where each set includes five images with the same digit label but different backgrounds and different styles of hand writing. Here, we adopt the terminology used by authors in Sutter et al. (2021), where they refer to each background as a "modality" to capture source specific information. Thus, for our experiments, each digit represents the shared information across modalities and different background images represent modality-specific information. In total we used for each digit and modality combination 1000 samples of training and validation examples (50000 images in total for ten digits and five modalities) and 891 samples of test examples (44550 images in total) from the original train and test split of the dataset. Our objective is to perform multi-class classification of ten digits across five different modalities, as shown in Figure 1(a).

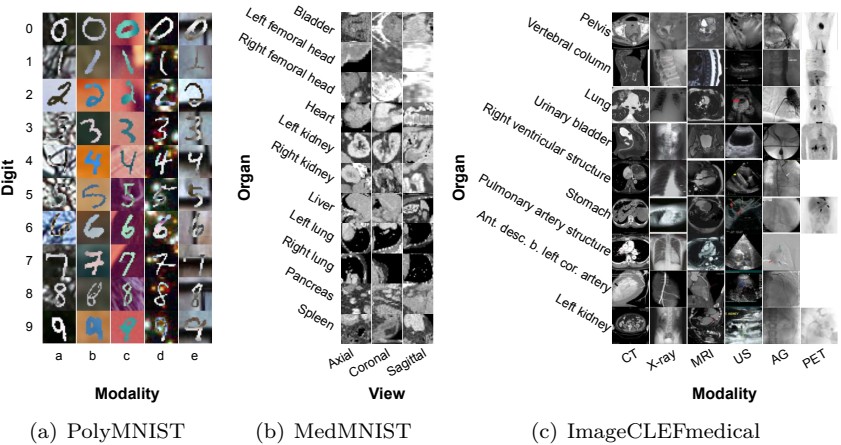

(a) PolyMNIST      (b) MedMNIST      (c) ImageCLEFmedical

Figure 1: We employ (a) PolyMNIST, multi-modal dataset for digit classification using data from different modalities, (b) MedMNIST for the classification of organs from different views of CT image slices, and (c) ImageCLEFmedical for organ classification using data from different imaging modalities.

### 2.1.2 MedMNIST

We use MedMNIST v2 Yang et al. (2023) benchmark to explore generalization across viewpoints. MedMNIST v2 is a large-scale MNIST-like dataset collection of standardized biomedical images, including datasets of 2D and 3D data. Among these, we use Organ{A,C,S}MNIST subset, which are based on CT images from *axial, coronal* and *sagittal* views. The visible organs within this data include *bladder, left femoral head, right femoral head, heart, left kidney, right kidney, liver, left lung, right lung, pancreas* and *spleen*. We used the original data split, with 61521 training (34581 for axial, 13000 for coronal and 13940 for sagittal view), 11335 validation (6491 for axial, 2392 for coronal and 2452 for sagittal view) and 34875 test (17778 for axial, 8268 for coronal and 8829 for sagittal view) samples. The goal is to perform multi-class classification of 11 body organs from axial, coronal and sagittal views. Examples of organ and view combinations are shown in Figure 1(b). Furthermore, number of samples for each combination are shown in Figure 2(a) and Table B, with (i) representing the training, (ii) validation, and (iii) test set.

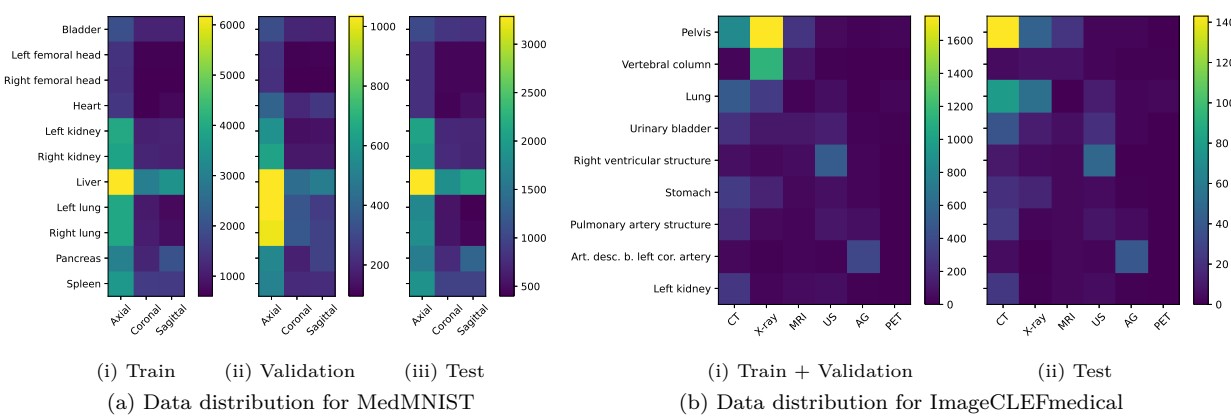

(i) Train    (ii) Validation    (iii) Test       (i) Train + Validation      (ii) Test

(a) Data distribution for MedMNIST      (b) Data distribution for ImageCLEFmedical

Figure 2: Number of images for medical datasets. (a) For MedMNIST: (i) training, (ii) validation, and (iii) test set. (b) For ImageCLEFmedical: (i) training and validation, and (ii) test set.

### 2.1.3 ImageCLEFmedical

We use ImageCLEFmedical Caption challenge Ionescu et al. (2022) dataset, a subset of the extended Radiology Objects in COntext (ROCO) dataset (Pelka et al., 2018). This dataset is derived from biomedical articles within the PMC OpenAccess subset, a comprehensive collection of figures sourced from open access biomedical journal articles (PubMed Central), along with radiology images extracted from original medical cases. In both training and validation data, each image is paired with Unified Medical Language System(UMLS) 2020 AB concepts (Bodenreider, 2004). These concepts represent UMLS terms, recognized as as CUIs (Concept Unique Identifiers) and extracted from the accompanying image captions. For instance, if the image caption contains terms such as "plain x-ray" or "pelvis", these concepts would be denoted for that image by the CUIs C1306645 and C0030797. Note that the ImageCLEFmedical Caption challenge comprises two subtasks: concept detection and concept prediction. Our experiments utilize the concept detection data within this dataset. There are more than 8000 concepts in the dataset, each with varying frequencies of occurrence. To enhance control and comprehension of generalization, we opted to work with a subset of images and concepts. Our analysis focused on the 100 most frequently employed CUIs. Filtering was then applied to images based on the semantic types of their associated concepts, specifically targeting concepts related to "Diagnostic procedure" for imaging modality identification, and "Body Part, Organ, or Organ Component" for presence of specific organs in the images. From the filtered list of CUIs, we selected a subset of organs for further analysis. This subset comprised nine distinct body organs, namely *pelvis, vertebral column, lung, urinary bladder, right ventricular structure, stomach, pulmonary artery structure, anterior descending branch of the left coronary artery,* and *left kidney.* As for imaging modalities, we considered *CT, X-ray, MRI, US, angiogram (AG),* and *PET* images. It's important to note that not all body parts are captured through all imaging modalities. Since the test images in the original dataset do not come with their concepts, we employed the train split from the original challenge dataset for training and validation, while the validation split was repurposed as the test set. This resulted in a dataset of 8433 images for training and validation, with an additional 688 images reserved for testing. Our task entails multi-class classification of the nine body organs across six distinct imaging modalities. Examples of body organ and modality combinations are shown in Figure 1(c). Furthermore, the number of samples for each combination are visualized in Figure 2(b) and Table C, where (i) illustrates the combined train and validation, and (ii) the test set.

## 2.2 Dataset specific tasks and domains

In our experiments, we focus on classifying different digits for PolyMNIST or organs for the medical datasets across different data domains. The datasets can be visualized as a grid structure illustrated in Figure 1 and each square within this grid represents a unique task/domain combination. These combinations consist of digit/modality combination for PolyMNIST, organ/view combination for MedMNIST and organ/modality combination for ImageCLEFmedical. Each row corresponds to a class, whether it be a digit or an organ, while each column signifies a specific data domain, such as a view or a modality. Each cell within the grid includes all the images from a particular combination of class and domain.

## 2.3 Generating train and validation splits for data diversity

The objective of this work is to conduct a comparative analysis and gain insights into specialized and multi-domain models under conditions of data-limited and OOD scenarios. To accomplish this, we create different data subsets characterized by differing aspects and levels of data diversity. We base the generation of these partitions on two key factors: data availability and the level of OOD, which we will explain in more detail below. During the testing phase, we refrain from any additional data processing and exclusively employ the predefined test splits as provided within each dataset.

### 2.3.1 Amount of data

For PolyMNIST, we start by constructing train and validation splits with different data distributions. For each of the digit and modality combination in the set, we have access to 1000 samples. To introduce diversity to data distribution, we implemented diverse probability distributions for digit and modality combinations,

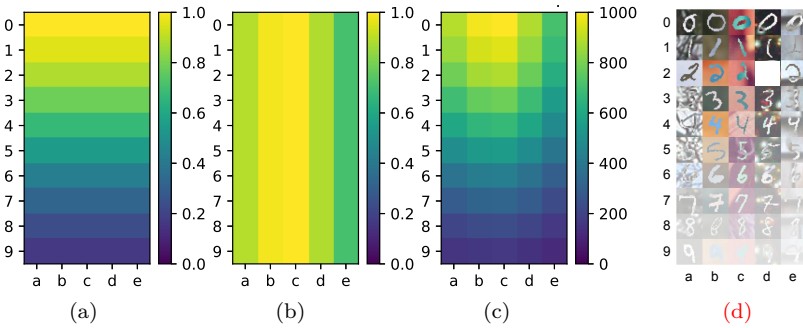

Figure 3: Data diversity evaluation for PolyMNIST: (a) normalized digit distribution using $\mu_{\text{digit}} = 0$ and $\sigma_{\text{digit}} = 5$, (b) normalized modality distribution with $\mu_{\text{modality}} = 2$ and $\sigma_{\text{modality}} = 3$, (c) number of samples from the resulting distribution for each digit/modality combination, (d) example of data distribution and OOD scenario with a 100% OOD level for digit *2* and modality *d*.

as follows. We use normal distribution for digit and modality distributions with the following parameters: the mean of the digit distribution, denoted as $\mu_{\text{digit}}$ is set to 0, and the standard deviation, $\sigma_{\text{digit}}$, takes values from the set $\{3, 5, 9, 17\}$. For the modality distribution, the mean, $\mu_{\text{modality}}$, is chosen from $\{0, 2\}$, while the standard deviation, $\sigma_{\text{modality}}$, is selected from $\{1, 3, 5\}$. We normalized these distributions, ensuring that all values are within the range of 0 to 1. We then multiply the digit and modality distributions and rescale the product by a factor of 1000, hereby guaranteeing that each distinct modality and digit combination has a sample count between 0 and 1000. In Figure 3(a-c), we present a graphical representation of this process, employing the specific parameter values of $\mu_{\text{digit}} = 0$, $\sigma_{\text{digit}} = 5$, $\mu_{\text{modality}} = 2$, and $\sigma_{\text{modality}} = 3$. The combination of different mean and standard deviation parameters for digit and modality distribution yields a total of 24 distinct distributions to use, as shown in Figure A.1.

Note that the datasets for both MedMNIST and ImageCLEFmedical inherently exhibit diversity, since they have varying sample counts for each organ across different views in MedMNIST and across different modalities in ImageCLEFmedical, as shown in Figure 2. Thus, we employ a range of sampling percentages to create distinct training and validation subsets for these datasets. We use their provided distributions and sample training and validation subsets accordingly. The sampling percentages include $\{5, 10, 25, 35, 50, 75, 100\}\%$, where 100% indicates the utilization of the entire available training and validation data, while 50% implies that only 50% of the data is incorporated. For instance, when employing a 100% sampling rate for MedMNIST, the multi-domain model utilized the complete training split consisting of 61521 images, 11335 images from the validation split, and 34875 images from the test split. At a 50% sampling percentage, the model then utilized 50% of the total available training and validation samples, resulting in 30760 samples from the training set, 5668 samples from the validation set, and retained the full 34875 test samples for evaluation. As a result, this sampling approach ensures consistent ratios of training and validation samples across different combinations, albeit with varying sample sizes.

### 2.3.2 Out-of-distribution (OOD) level

To evaluate the OOD performance of both specialized and multi-domain models across various datasets, we introduce *OOD levels*, as follows: for each task and data domain combination, we systematically exclude a subset of instances from both the training and validation sets. Subsequently, we repeat the training and validation procedures for each of the combination.

We quantify OOD levels using percentages, specifically $\{0, 25, 50, 75, 85, 95, 100\}\%$, where 100% signifies that the specific combination is entirely absent from both the training and validation sets, whereas, 0% indicates that the training and validation datasets contain the complete set of samples for the given combination. As a result, for each experiment, the specific combination becomes either never or less frequently observed. For a fair representation of each combination, we ensure that every combination occurs exactly once, that is, each row and column features only a single cell representing a specific combination.

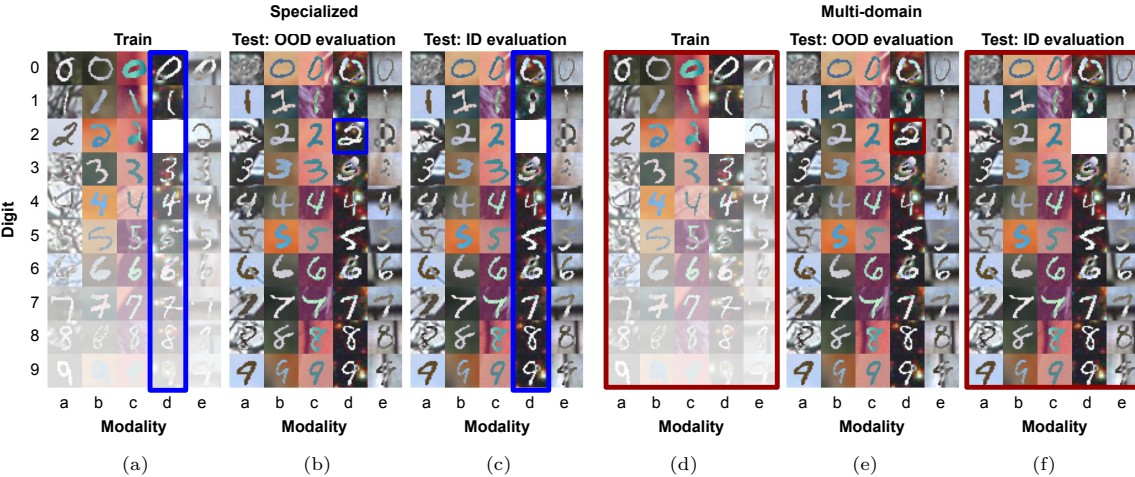

Figure 4: Training and evaluation schemes for specialized and multi-domain models: (a,d) represent the training data utilized by specialized and multi-domain models using $\mu_{\text{digit}} = 0$, $\sigma_{\text{digit}} = 5$, $\mu_{\text{modality}} = 2$ and $\sigma_{\text{modality}} = 3$ with a 100% OOD level for digit $2$ and modality $d$. (b,e) show the OOD evaluation specifically for digit $2$ and modality $d$. (c,f) demonstrate the ID evaluation for all other digit/modality combinations except digit $2$ and modality $d$.

We present an illustrative example of diverse data strategies for PolyMNIST in Figure 3(d). Here, opacity is used to represent the quantity of data, with less opacity indicating a larger amount of data. Specifically, we employed distribution parameters $\mu_{\text{digit}} = 0$, $\sigma_{\text{digit}} = 5$, $\mu_{\text{modality}} = 2$, and $\sigma_{\text{modality}} = 3$ to define the data distribution, as in Figure 3(c). Furthermore, it also showcases the *OOD scenario* with a 100% OOD level for digit $2$ and modality $d$, meaning the model will never see the digit of $2$ from the modality $d$.

## 2.4 Training and testing procedure

In our experimental setup, we employ same datasets for both specialized and multi-domain models. The key distinction lies in the manner with which (part of) data the models are trained. To elaborate, for instance, in the case of PolyMNIST, we train and evaluate five distinct specialized models, each dedicated to classifying digits based on five modalities using the respective data from each modality. An example for modality-specific specialized models for the example data in Figure 3(d) is shown in Figure 4(a). In contrast, the multi-domain model leverages the entire available data for the same digit classification task, as shown in Figure 4(d). It's important to clarify that the choice between a multi-domain or specialized setup impacts the training and validation phases, determining which data partitions the models are exposed to. When it comes to the test phase, we assess each test image to predict its designated task. Therefore, the multi-domain doesn't require the simultaneous input of all modalities during testing, e.g. in contrast to multi-modal learning.

# 3 Experiments and Results

## 3.1 Experimental Setup

### 3.1.1 Model Architecture and Hyperparameters

In our experiments, we utilize the pre-trained ResNet-18 architecture (He et al., 2016), employing the cross-entropy loss and the AdamW optimizer (Loshchilov & Hutter, 2017). PolyMNIST and ImageCLEFmedical training data is split into training and validation sets with a ratio of 0.75. For testing we use the validation data. For MedMNIST, we use the official train/validation/test splits. For MedMNIST and ImageCLEFmedical datasets, we evaluate and report the average of five random seeds. We train the models for 25 epochs and decay the learning rate by 0.1 every 5 epochs. For PolyMNIST, we set the learning rate to 0.005, use

a batch size of 512 and employ a weight decay of 0.001. For MedMNIST, the learning rate is set to 0.001, batch size to 128, weight decay to 0.001. For ImageCLEFmedical, we use a learning rate of 0.0005, utilize a batch size of 128, set the weight decay to 0.00001. Note that, for hyperparameter tuning, we employed multi-domain models and conducted a grid search for optimizing learning rate, weight decay and different ResNet architectures such as ResNet-18, ResNet-34 and ResNet-50. We then repeated this for specialized models, with the hyperparameters largely aligned, ensuring that they did not affect the training convergence. The images in PolyMNIST and MedMNIST have the resolution of $28 \times 28$ pixels$^2$. We pre-process these images by resizing them to $32 \times 32$ pixels$^2$. As for ImageCLEFmedical, we center crop the images to ensure equal width and length, further augmenting them with a random 0x to 0.1x translation and resizing them to dimensions of $224 \times 224$ pixels$^2$.

### 3.1.2 Evaluation

We employ balanced accuracy as our evaluation metric, encompassing two distinct evaluation scenarios: out-of-distribution (OOD) accuracy and in-distribution (ID) accuracy. This involves evaluating the accuracy of each excluded combination and subsequently computing the average accuracy across all such combinations. For PolyMNIST, this approach results in a total of 50 evaluations (10 digits and 5 modalities), while for MedMNIST, we conduct 33 evaluations (11 organs and 3 views), and for ImageCLEFmedical, there are 54 evaluations (9 organs and 6 modalities) in total. We refer to the resulting metric as OOD average balanced accuracy. In addition, we calculate the average accuracy of all combinations except for the excluded ones and once again calculate the average of all the different combinations set. We designate this outcome as ID average balanced accuracy. All evaluation metrics are reported in test set. As an example for PolyMNIST, the difference between each of OOD and ID evaluation for both specialized and multi-domain models are shown in Figure 4. Specifically, (a-c) illustrate the specialized setup, while (d,f) the multi-domain setup, featuring a 100% OOD level for digit *2* and modality *d*, ensuring that the models have no exposure to cases involving digit *2* and modality *d*. For the OOD evaluation, (b, e), both models are evaluated on test samples featuring digit *2* from modality *d*. For the ID evaluation, (c, f), the specialized model is assessed using test samples from modality *d* and all digits except *2*, while the multi-domain model undergoes evaluation for all other digit/modality combinations except digit *2* and modality *d*. Note that, in our work average balanced accuracy corresponds to the mean accuracy across all task/domain combinations, which is different from overall accuracy or balanced accuracy. Our evaluation approach involves a detailed breakdown based on task and domain.

## 3.2 Results

### 3.2.1 PolyMNIST

We begin our analysis by assessing data diversity across various OOD levels. For this, we not only compare specialized and multi-domain models, but but also introduce a modified version of specialized models, which we refer to as *specialized upsampled models*: we augment the data available to specialized models, ensuring that each digit is classified with an equal number of images for both the specialized upsampled and multi-domain models. The original PolyMNIST dataset provides a sufficient number of samples for this purpose. Thus, both specialized and multi-domain models have access to a maximum of 1000 images for each digit/modality combination, while the upsampled counterparts of the specialized models benefit from an expanded dataset, containing up to 5000 images for each such combination. Furthermore, we would like to mention that the scenario involving specialized upsampled models is not a realistic representation but is exclusively examined to assess the advantages and limitations associated with an augmented dataset.

Figure 5 compares the performance of specialized, specialized upsampled and multi-domain models. We compute the area under the curve (AUC) for both the OOD and ID average balanced accuracy curves across various data distributions. Each data point represents the AUC for different OOD level, where (a) provides a comparison of ID average balanced accuracy across different data distributions and OOD levels, while (b) shows OOD average balanced accuracy among specialized (blue), specialized upsampled (green), and multi-domain (red) models. Note that, in this experiment, the highest achievable AUC for a model is marked with a dashed black line. This is calculated using the highest possible average balanced accuracy, which can reach

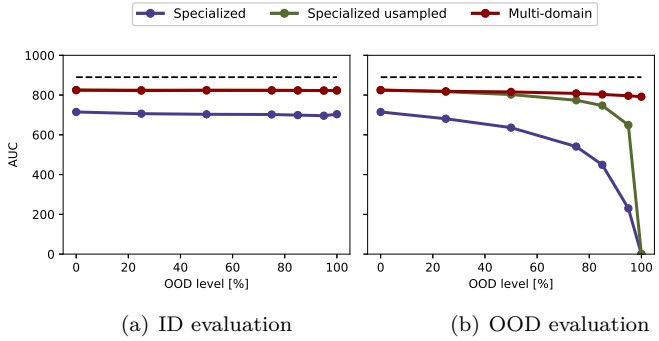

(a) ID evaluation      (b) OOD evaluation

Figure 5: Evaluating OOD levels for PolyMNIST. Each point shows the area under the balanced accuracy curve through different evaluation of data distributions for different OOD levels in x-axis.

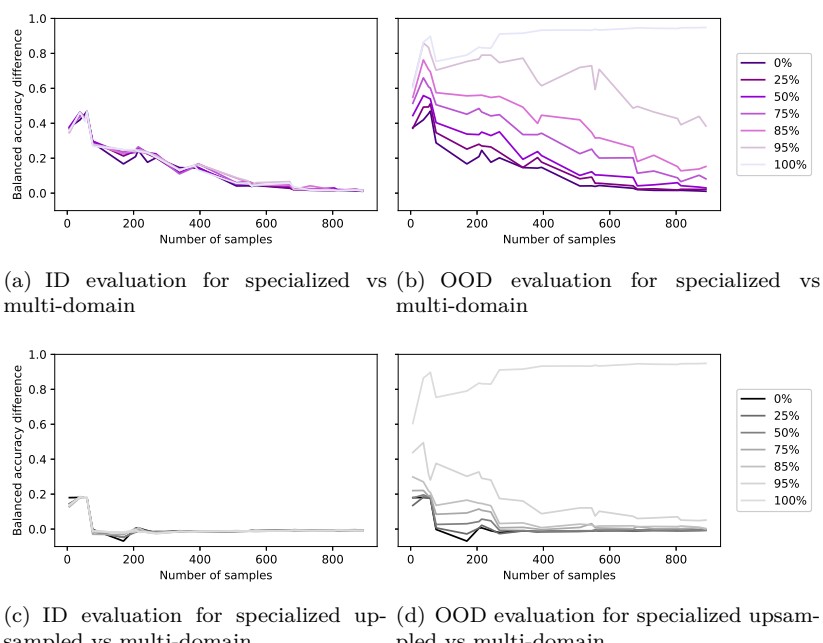

(a) ID evaluation for specialized vs multi-domain

(b) OOD evaluation for specialized vs multi-domain

(c) ID evaluation for specialized up-sampled vs multi-domain

(d) OOD evaluation for specialized upsampled vs multi-domain

Figure 6: Evaluating amount of data for PolyMNIST. Reporting balanced accuracy difference between specialized and multi-domain models (a,b) and between specialized upsampled and multi-domain models (c,d). Each line corresponds to different OOD levels and presents the difference between models across various data distribution evaluations, as indicated on the x-axis.

1, and the total number of samples, maximum of 890 (as shown in Figure A.1). Consequently, the dashed line reflects a maximum AUC of 890. The ideal outcome is represented by a flat line at maximum, indicating perfect performance unaffected by OOD levels. Figure 6 provides an in-depth overview of the models across diverse data distributions. To facilitate a meaningful comparison across different distributions, we have organized the 24 distributions in ascending order based on their median values, as shown in Figure A.1. The datapoints on x-axis represent these 24 different data distributions, labeled as the titles of each subfigure in Figure A.1. They span from a minimum of 6 (median of the distribution in the first row, first column) to a maximum of 890 (median of the distribution in the second row, last column). We depict the balanced accuracy difference between specialized and multi-domain models (a,b) and specialized upsampled and multi-domain models (c,d). Each line represents OOD levels showing the differences between models during data distribution evaluations. For a more detailed overview, please refer to Figure 7, which provides an overview of the average balanced accuracy scores for specialized (a,b), specialized upsampled (c,d) and multi-domain

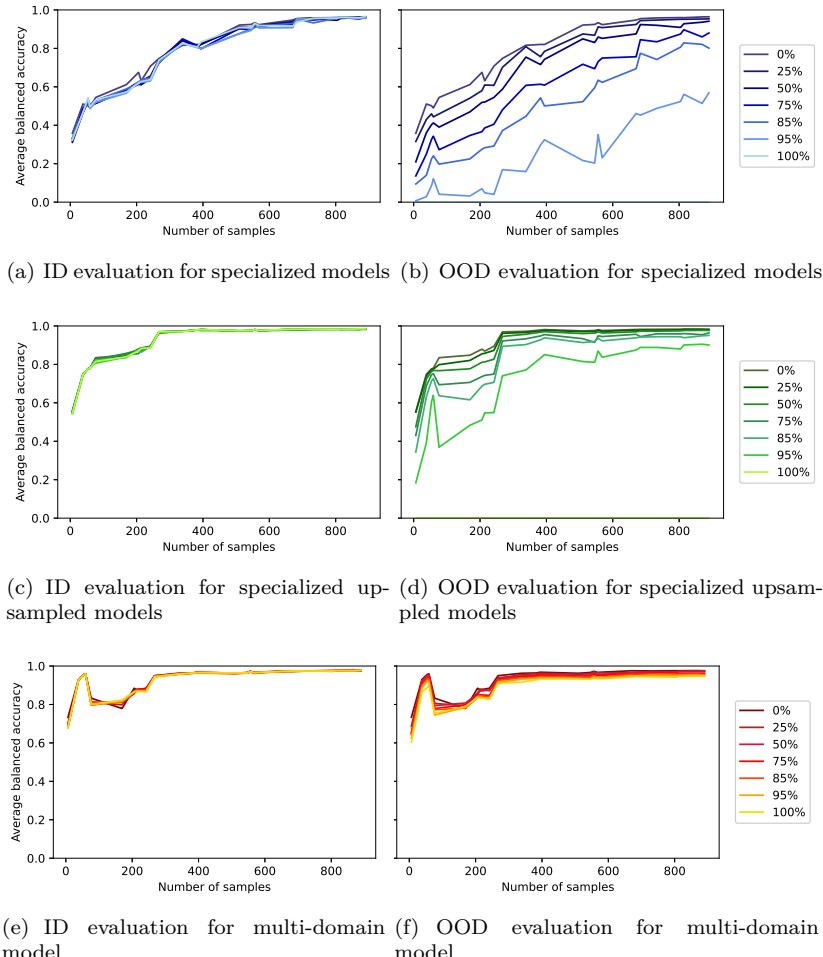

(a) ID evaluation for specialized models

(b) OOD evaluation for specialized models

(c) ID evaluation for specialized up-sampled models

(d) OOD evaluation for specialized upsampled models

(e) ID evaluation for multi-domain model

(f) OOD evaluation for multi-domain model

Figure 7: PolyMNIST across various data distributions. We report average balanced accuracy for specialized (a,b), specialized upsampled (c,d) and multi-domain (e,f) models across various distributions, depicted in x-axis, and OOD levels, indicated by different color codes, for both ID (a,c,e) and OOD (b,d,f) evaluation.

(e,f) models. These are presented across various distributions on the x-axis, and OOD levels, indicated by different color codes for both ID (a,c,e) and OOD (b,d,f) evaluation. Each data point in Figure 7 represents the average accuracy obtained from 50 distinct experiments. In these experiments, one digit (out of 10 digits: 0 to 9) and one modality (out of 5 modalities: a to e) were systematically excluded from training and validation, and this process was repeated 50 times to encompass all possible task/modality combinations. Figure 5 was then calculated as the the AUC of each line representing the OOD levels in these figures. Specifically, in Figure 5(a) and (b), the AUC values represented by the blue lines correspond to the areas under the curves for different OOD levels illustrated in Figure 7(a) and (b). Similarly, for the red lines in Figure 5(a) and (b), each calculated AUC value corresponds to the area under different OOD curves shown in Figure 7(e) and (f). Additionally, we have computed the balanced accuracy difference for each OOD level in these figures, with multi-domain minus specialized models, and the outcomes are presented in Figure 6.

When assessing the ID accuracy, our results indicate that varying levels of OOD scenarios do not notably impact ID accuracy. The performance of the specialized models are still lower than the specialized upsampled and multi-domain models, showing that ID accuracy can be compensated with higher number of samples. Upon analyzing OOD performance, a notable and consistent pattern emerges: as the OOD level increases, OOD accuracy declines noticeably for specialized and specialized upsampled models. This stands in stark contrast to multi-domain models, which exhibit considerably greater resilience to this phenomenon. The

difference becomes particularly pronounced for OOD levels >50%. This can be attributed to the fact that multi-domain models benefit from shared information across different modalities for the classification task, thereby aiding OOD recovery. Specifically, we are referring to digit specific information as part of the shared information. In contrast, specialized models struggle to recover unseen (at OOD level 100%) or scarce encountered (at OOD level < 100%) digit/modality combinations, even when provided with larger sample sizes with specialized upsampled models. Furthermore, across all models, we observe a consistent trend: both ID and OOD performance declines as the number of samples decreases. In Figure 7(e) and (f), the accuracy does not exhibit a monotonic increase; instead, there is a peak at a smaller number of samples. We suspect that this peak at the small sample size is caused by the double descent phenomenon. This is because while the training error and loss function decrease in our experiments, the test error increases with a small number of samples.

In experiments, specialized models were trained exclusively with data from their corresponding modalities. For instance, a specialized model for modality $d$ is exclusively trained using data from modality $d$ and evaluated on modality $d$, as illustrated in Figure 3(i-iii). We further evaluated a scenario where the specialized model, initially trained on data from modality $d$, was tested on other modalities $a, b, c, e$ to evaluate the robustness of the learned representations of the specialized models. For this we evaluated different OOD levels for digit $2$ and modality $d$. In summary, training on one domain and testing on another leads to large drops of accuracy, showing that the specialized models have limited generalization capabilities to other domains. More specifically, for OOD level of 50%, the average balanced accuracy evaluated on digit $2$ and domain $d$ yielded 0.95, whereas on domains $a,b,c,e$ yielded 0.71. Average balanced accuracy tested on all digits except digit $2$ and on modality $d$ yielded 0.94, in contrast, it was 0.77 for the modalities $a,b,c,e$. For OOD level of 85%, the average balanced accuracy evaluated on digit $2$ and domain $d$ yielded 0.87, whereas on domain $a,b,c,e$ it was 0.59. Average balanced accuracy tested on all digits except digit $2$ and on modality $d$ was 0.94, in contrast it was 0.73 on domains $a,b,c,e$. In contrast, the multi-domain model learns a single model across all domains, is more robust to different domains, which results in OOD generalization, as shown in Figures 6(b) and 7(f).

For testing potential knowledge transfer, we conducted another control experiment, where our goal was to evaluate a scenario where information sharing is constrained for multi-domain model. For this, we split the data from various modalities into two distinct domains, grouping classes as follows: 0 and 5 together as one class, 1 and 6 as another, and so on, with classes 4 and 9 comprising the final group. Thus, we split digits 0 through 4 into one domain and digits 5 through 9 into another. Consequently, our multi-domain model exploits both domains, with each class encompassing two dissimilar digits. We then run OOD level experiments for each digit and repeat this experiment for each of the original modalities $a$ to $e$. In Figure 8, we present the OOC evaluation results, showcasing the average balanced accuracy achieved by aggregating the experiments. The lower the OOD level, the higher is the accuracy. However, when the OOD level reaches 100%, the average accuracy declines to a level expected by chance. This observation shows the fact that at 100% OOD, there is no opportunity for knowledge transfer, emphasizing the absence of information sharing.

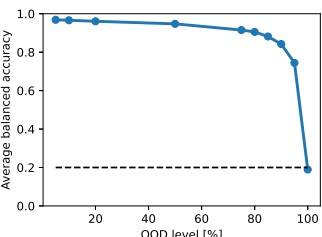

Figure 8: Control experiment for PolyMNIST. We test the potential knowledge transfer and evaluate a scenario where information sharing is limited for multi-domain model. Dashed line shows the level of chance.

### 3.2.2 MedMNIST

In Figure 9, we present a comparison of AUC values achieved by specialized and multi-domain models across various sampling percentages for both ID (a) and OOD (b) average balanced accuracy curves. The dashed black line, marked with a value of 100, represents the highest attainable AUC, which corresponds to a 100% sampling rate and perfect accuracy. Figure 10 provides a more comprehensive analysis of the specialized and multi-domain models with different amount of data for both ID evaluation (a) and OOD evaluation (b). Each line corresponds to different OOD levels and illustrates the differences between the models across different sampling percentages, as indicated on the x-axis. For a detailed overview, please refer to Figure B.1, which presents an overview of the average balanced accuracy scores for specialized and multi-domain models. Furthermore, in Figure B.2, Figure B.3 and B.4, we report the AUC, accuracy differences, and model accuracies at view level.

When assessing the ID accuracy, such as in the case of PolyMNIST, our findings suggest that varying OOD levels do not significantly impact ID accuracy and both models exhibit similar levels of accuracy. For the OOD performance, as the OOD level increases, OOD accuracy experiences a noticeable decline for both models. This distinction between the models becomes particularly pronounced for OOD levels exceeding 75%. In the extreme case of a 100% OOD level, the specialized model's accuracy drops, which makes it impossible for specialized models to predict fully unseen data. In contrast, the multi-domain model still benefits from shared information in this scenario, namely the organ specific information across views. Furthermore, across all models, both ID and OOD performance decrease as the number of samples for each digit/modality combination decreases.

Figure 2(a) displays the image distribution across organ/view combinations in MedMNIST's train/validation/test splits. Across all splits, the axial view consistently contains the highest number of

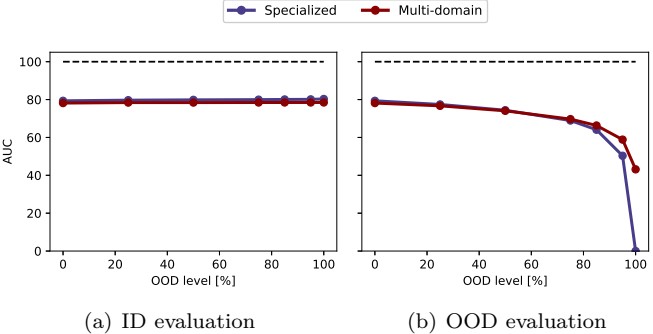

(a) ID evaluation       (b) OOD evaluation

Figure 9: OOD levels for MedMNIST. Each point shows the area under the balanced accuracy curve through different evaluation of data availability (sampling percentage) for different OOD levels, shown in x-axis.

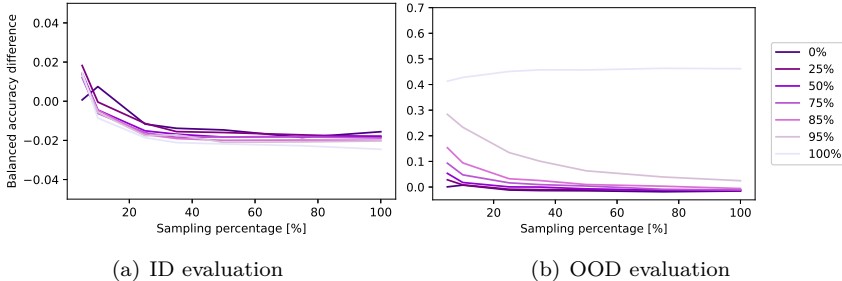

(a) ID evaluation       (b) OOD evaluation

Figure 10: Amount of data for MedMNIST. Each line corresponds to different OOD levels and presents the balanced accuracy difference between specialized and multi-domain models across different amount of data (sampling percentage), as indicated on the x-axis.

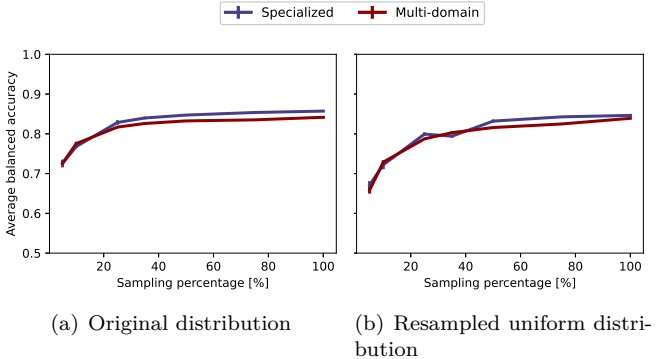

(a) Original distribution  (b) Resampled uniform distribution

Figure 11: Evaluating data distribution in MedMNIST. ID balanced accuracy for specialized and multi-domain models across different amount of data. Results are presented for two scenarios: (a) utilizing the official data split and (b) employing a resampled uniform distribution.

images, notably with the liver in the axial view having the most images in both the train and test splits. To investigate whether the matching ID accuracy between specialized and multi-domain models can be attributed to data distribution, we conducted a control experiment. For this, we restructured the data by resampling so that each organ/view combination contained 600 samples for training and 95 samples for the validation split, aligning with the minimum sample size observed in the official training and validation split. We repeat the experiment with using the resampled dataset and for different amount of data using sampling percentage. As an example, with a 50% sampling percentage, each organ/view combination benefits from 300 training and 48 validation samples. Figure 11 provides a comparison of these different distributions for OOD level of 0%. These show that the data distribution has a negligible impact on ID accuracy, as the resampled uniform distribution data continues to demonstrate matching ID accuracy between specialized and multi-domain models.

### 3.2.3 ImageCLEFmedical

In Figure 12, we compare the models in terms of their AUC values under ID (a) and OOD (b) average balanced accuracy curves. Figure 13 presents a more in-depth comparison of the specialized and multi-domain models in terms of varying amount of data for both ID (a) and OOD (b) evaluation. For a detailed overview, please refer to Figure C.1, which provides a summary of the average balanced accuracy scores for specialized and multi-domain models.

For ID accuracy, our findings are consistent with the outcomes observed in PolyMNIST and MedMNIST studies. Here, variations in OOD scenarios do not significantly impact ID accuracy. In contrast to MedMNIST,

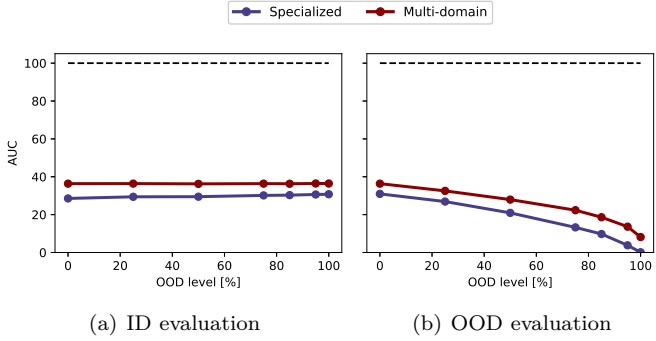

(a) ID evaluation  (b) OOD evaluation

Figure 12: OOD levels for ImageCLEFmedical. Each point shows the area under the balanced accuracy curve through different data availability (sampling percentage) for different OOD levels, shown in x-axis.

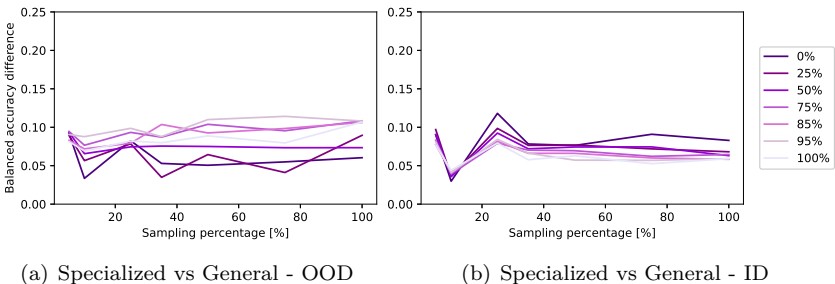

(a) Specialized vs General - OOD   (b) Specialized vs General - ID

Figure 13: Amount of data for ImageCLEFmedical. Each line corresponds to different OOD levels and presents the balanced accuracy difference between specialized and multi-domain models across different amount of data (sampling percentage), as indicated on the x-axis.

the multi-domain model demonstrates an 8% improvement in accuracy, mirroring the findings in PolyMNIST where multi-domain model outperformed specialized models. This result is particularly intriguing given that the utilized images are unprocessed and have remarkable diversity. Regarding OOD performance, as the OOD level increases, both models experience a noticeable decline. Importantly, the multi-domain model maintains a consistent 8% advantage across all OOD levels, where the multi-domain model continues to benefit from shared information for both ID and OOD evaluation, even in data-limited and OOD scenarios.

## 4 Discussion and Conclusion

Motivated by the recent advancements in foundation models exploiting diverse data sources and demonstrating exceptional generalization abilities, this work introduced a multi-domain strategy and compared its performance with the single domain specialized approach, particularly within the context of medical image analysis, where the latter is the norm. These are evaluated in scenarios involving OOD and data-limited scenarios using three datasets, such as the toy dataset PolyMNIST (Sutter et al., 2021), as well as two medical datasets, MedMNIST (Yang et al., 2023) and ImageCLEFmedical (Ionescu et al., 2022) and obtain following key conclusions:

- Multi-domain models outperform specialized models in OOD and data-limited scenarios, capitalizing on their ability to leverage shared information across diverse domains.

- Multi-domain models consistently either match or excel specialized models in terms of their ID accuracy.

- Specialized models can compensate for ID accuracy with a higher number of samples. However, they face considerable challenges in recovering OOD accuracy for tasks that are entirely unseen or encountered only infrequently, even when provided with larger sample sizes.

- The level of OOD scenario does not impact ID accuracy for any of the models, indicating the robustness in preserving ID accuracy across varying OOD levels.

Based on our findings, the advantage of multi-domain model is particularly pronounced for OOD tasks when OOD level exceeds 80%. An OOD level surpassing 80% highlights instances where the model encounters class/domain combinations either extremely rarely or never before. Such scenarios are common in medical applications, particularly those involving rare diseases or conditions. Accessing medical conditions from diverse data domains could be beneficial, particularly when a specific condition hasn't been frequently observed within a particular domain in the training data. It's worth noting that the extent of the advantage of knowledge transfer between domains is limited upon the availability of shared information.

As a future direction, understanding the underlying mechanisms behind the generalization capabilities of multi-domain models for OOD and data-limited scenarios is a crucial direction. Deeper investigations into

the information sharing within these models hold the potential to yield more efficient strategies for knowledge transfer and domain adaptation. Moreover, a potential future work is to improve multi-domain models by incorporating domain-specific knowledge into the training process (Li et al., 2020a; Ahn et al., 2020; Xie et al., 2021a; Guan & Liu, 2022), e.g. through the use of alternative loss functions. Furthermore, addressing the scalability of these models for real-world, large-scale applications remains a pressing concern for medical image analysis. Future research can concentrate on working with refining these models for efficiency and ensuring their practicality in real-world resource-constrained environments. Additionally, the exploration of more diverse datasets and problem domains will be essential for validating and extending our findings.

In summary, our work underlines the effectiveness of multi-domain models in tackling OOD and data-limited challenges, offering promising avenues for their application in medical image analysis where such challenges are prevalent. These insights contribute to the ongoing exploration and implementation of large scale models in diverse fields and applications.

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

# A PolyMNIST

In Figure A.1, we present 24 distinct data distributions, each representing the number of samples within training and validation splits. To be able to summarize and visualize these, we calculated the median of each of the 24 distributions. These values are displayed in the titles of each subfigure in Figure A.1 and correspond to the x-axis values for the evaluations in Figure 6 and Figure 7.

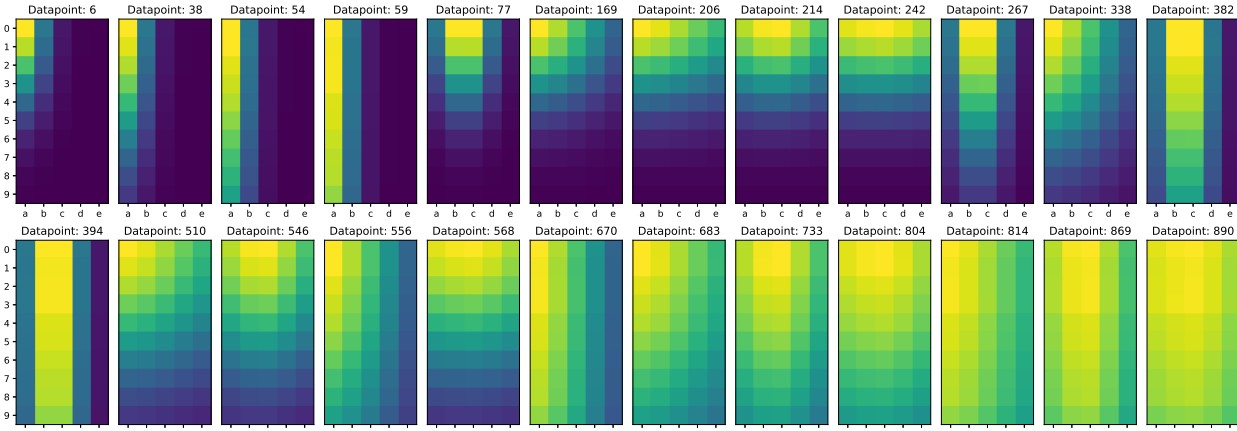

Figure A.1: 24 distinct data distributions, each representing the number of samples within various training and validation splits. These are characterized by diverse probability distributions for digit and modality combinations. They are organized in ascending order, based on their respective median values.

We conducted an additional experiment, mirroring the amount of data experiments conducted with MedMNIST and ImageCLEFmedical datasets using the sampling percentage. For this, we used a uniform sample distribution for each digit/modality combination having 1000 samples. Subsequently, we performed sampling at rates of $\{5, 10, 25, 35, 50, 75, 100\}\%$, resulting further in a uniform distribution. For example, when using a 10% sampling percentage, we obtained 100 samples for each digit/modality combination. Figure A.2 reports AUC under the average balanced accuracy curves across sampling percentage for various OOD levels for PolyMNIST. Notably, these results underscores the similar trend to those observed in Figure 5.

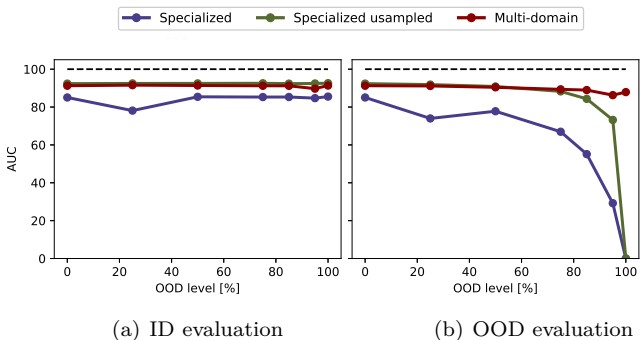

(a) ID evaluation          (b) OOD evaluation

Figure A.2: OOD levels for PolyMNIST. Each point shows the area under the balanced accuracy curve through different evaluation of sampling percentage for different OOD levels, shown in x-axis.

## B  MedMNIST

Table B displays the distribution of images in the MedMNIST dataset categorized by organs including *bladder, left femoral head, right femoral head, heart, left kidney, right kidney, liver, left lung, right lung, pancreas* and *spleen*, as well as by *axial, coronal,* and *sagittal* views.

| | (i) Train | | | (ii) Validation | | | (iii) Test | | |
|---|---|---|---|---|---|---|---|---|---|
| **Organ** | **Axial** | **Coronal** | **Sagittal** | **Axial** | **Coronal** | **Sagittal** | **Axial** | **Coronal** | **Sagittal** |
| Bladder | 1956 | 1153 | 1148 | 321 | 191 | 188 | 1036 | 833 | 811 |
| Left femoral head | 1408 | 626 | 637 | 233 | 102 | 104 | 784 | 442 | 439 |
| Right femoral head | 1359 | 608 | 615 | 225 | 96 | 95 | 793 | 441 | 447 |
| Heart | 1474 | 600 | 721 | 392 | 202 | 246 | 785 | 421 | 510 |
| Left kidney | 3963 | 1088 | 1132 | 568 | 132 | 140 | 2064 | 732 | 704 |
| Right kidney | 3817 | 1170 | 1119 | 637 | 157 | 159 | 1965 | 737 | 693 |
| Liver | 6164 | 2986 | 3464 | 1033 | 429 | 491 | 3285 | 1836 | 2078 |
| Left lung | 3919 | 1002 | 741 | 1033 | 347 | 261 | 1747 | 550 | 397 |
| Right lung | 3929 | 1022 | 803 | 1009 | 352 | 275 | 1813 | 558 | 439 |
| Pancreas | 3031 | 1173 | 2004 | 529 | 179 | 280 | 1622 | 750 | 1343 |
| Spleen | 3561 | 1572 | 1556 | 511 | 205 | 213 | 1884 | 968 | 968 |

Table B.1: Number of images for MedMNIST dataset for (i) training, (ii) validation, and (iii) test set.

Figure B.1 presents a comprehensive summary of the average balanced accuracy scores for both specialized and multi-domain models for different OOD levels and amount of data. We report the mean and standard deviation (as the error bar) of the test accuracy across five random seeds.

Figure B.2 reports the AUC, Figure B.3 highlights the accuracy differences, and Figure B.4 shows the accuracy of the specialized and multi-domain models at a more granular view level.

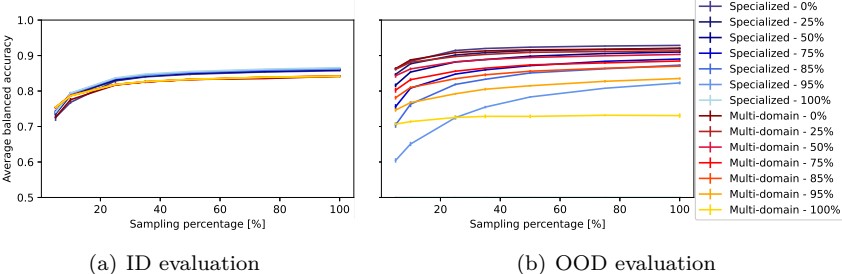

(a) ID evaluation          (b) OOD evaluation

Figure B.1: Comparison of models across various amount of data for MedMNIST. We report average balanced accuracy for specialized and multi-domain models across various sampling rates, depicted in x-axis, and OOD levels, indicated by different color codes, for both ID (a) and OOD (b) evaluation. We report the mean and standard deviation of the test accuracy across five random seeds.

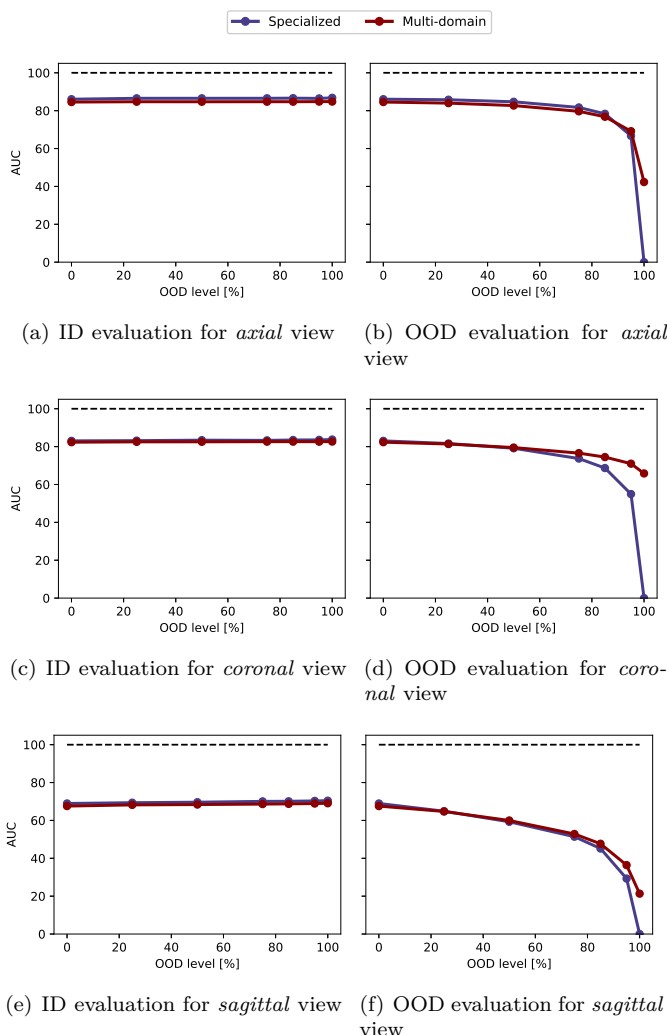

(a) ID evaluation for *axial* view

(b) OOD evaluation for *axial* view

(c) ID evaluation for *coronal* view

(d) OOD evaluation for *coronal* view

(e) ID evaluation for *sagittal* view

(f) OOD evaluation for *sagittal* view

Figure B.2: Evaluating OOD levels for MedMNIST for each of the views: axial (a,b), coronal (c,d), and sagittal (e,f). Each point shows the area under the balanced accuracy curve through different evaluation of data availability (sampling percentage) for different OOD levels, shown in x-axis.

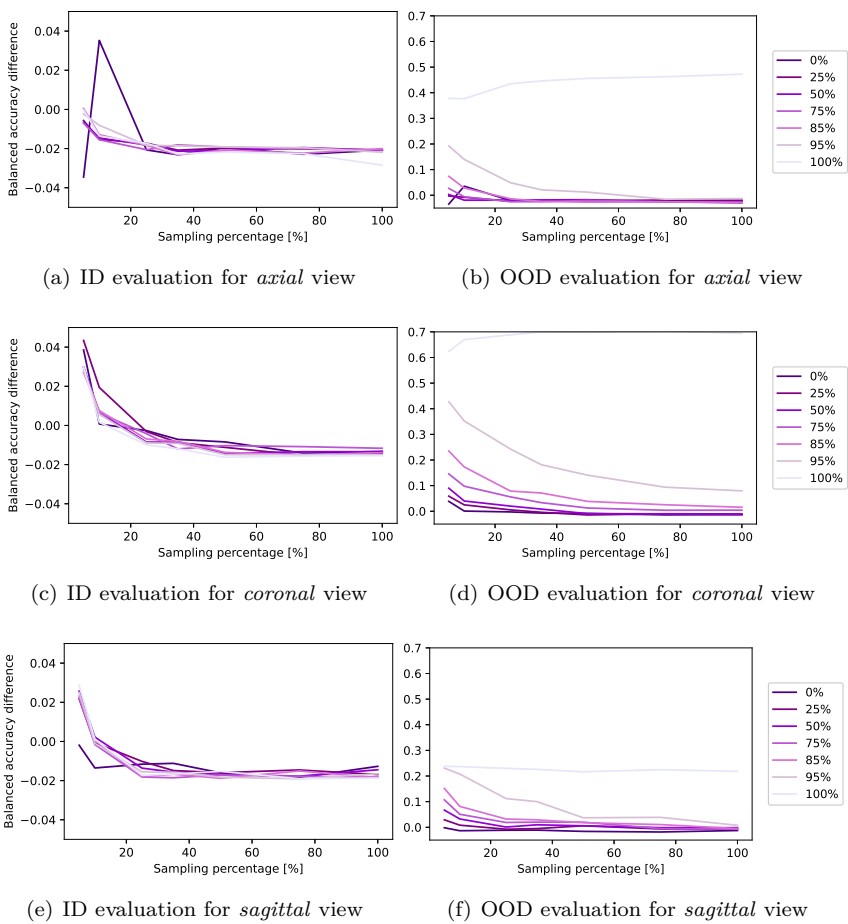

(a) ID evaluation for *axial* view     (b) OOD evaluation for *axial* view

(c) ID evaluation for *coronal* view     (d) OOD evaluation for *coronal* view

(e) ID evaluation for *sagittal* view     (f) OOD evaluation for *sagittal* view

Figure B.3: Evaluating amount of data for MedMNIST for each of the views: axial (a,b), coronal (c,d), and sagittal (e,f). Each line corresponds to different OOD levels and presents the balanced accuracy difference between specialized and multi-domain models across different amount of data (sampling percentage), as indicated on the x-axis.

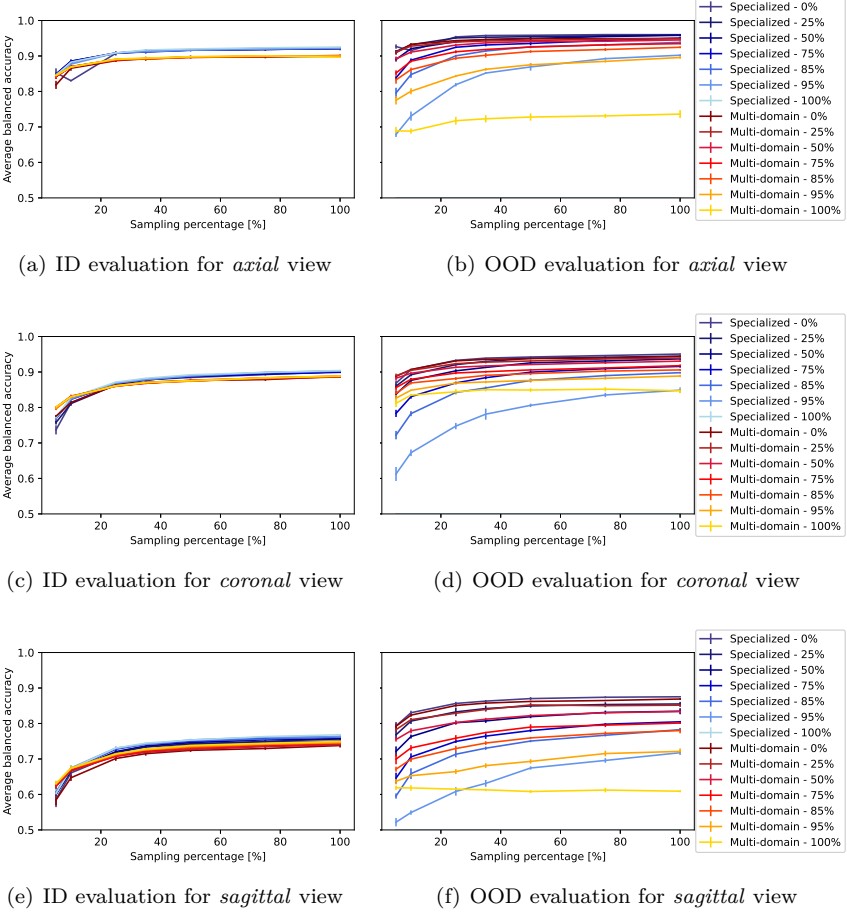

(a) ID evaluation for *axial* view

(b) OOD evaluation for *axial* view

(c) ID evaluation for *coronal* view

(d) OOD evaluation for *coronal* view

(e) ID evaluation for *sagittal* view

(f) OOD evaluation for *sagittal* view

Figure B.4: Comparison of models across various amount of data for MedMNIST for each of the views: axial (a,b), coronal (c,d), and sagittal (e,f). We report average balanced accuracy for specialized and multi-domain models across sampling rates, depicted in x-axis, and OOD levels, indicated by different color codes, for both ID (a,c,e) and OOD (b,d,f) evaluation. We report the mean and standard deviation of the test accuracy across five random seeds.

## C   ImageCLEFmedical

Table C displays the distribution of images in the ImageCLEFmedical dataset categorized by organs including *pelvis, vertebral column, lung, urinary bladder, right ventricular structure, stomach, pulmonary artery structure, anterior descending branch of the left coronary artery,* and *left kidney,* as well as imaging modalities *CT, X-ray, MRI, US, AG,* and *PET.*

| (i) Train + Validation | | | | | | |
|---|---|---|---|---|---|---|
| **Organ** | **CT** | **X-ray** | **MRI** | **US** | **AG** | **PET** |
| Pelvis | 826 | 1747 | 262 | 45 | 20 | 23 |
| Vertebral column | 23 | 1134 | 101 | 8 | 3 | 2 |
| Lung | 492 | 313 | 14 | 63 | 6 | 24 |
| Urinary bladder | 251 | 116 | 116 | 149 | 14 | 10 |
| Right ventricular structure | 70 | 36 | 59 | 507 | 28 | 0 |
| Stomach | 311 | 171 | 39 | 71 | 21 | 3 |
| Pulmonary artery structure | 219 | 42 | 24 | 109 | 74 | 0 |
| Ant. desc. b. left cor. artery | 35 | 7 | 23 | 20 | 373 | 0 |
| Left kidney | 275 | 29 | 51 | 64 | 8 | 2 |
| (ii) Test | | | | | | |
| **Organ** | **CT** | **X-ray** | **MRI** | **US** | **AG** | **PET** |
| Pelvis | 143 | 45 | 22 | 2 | 2 | 0 |
| Vertebral column | 4 | 7 | 7 | 2 | 1 | 1 |
| Lung | 79 | 53 | 0 | 11 | 2 | 3 |
| Urinary bladder | 37 | 11 | 6 | 19 | 2 | 0 |
| Right ventricular structure | 10 | 4 | 3 | 47 | 0 | 0 |
| Stomach | 19 | 15 | 3 | 4 | 1 | 0 |
| Pulmonary artery structure | 24 | 2 | 3 | 8 | 4 | 0 |
| Ant. desc. b. left cor. artery | 5 | 0 | 3 | 2 | 39 | 0 |
| Left kidney | 23 | 1 | 3 | 5 | 1 | 0 |

Table C.1: Number of images for ImageCLEFmedical dataset for (i) train and validation and (ii) test set.

Figure C.1 presents a comprehensive summary of the average balanced accuracy scores for both specialized and multi-domain models for different OOD levels and amount of data. We report the mean and standard deviation (as the error bar) of the test accuracy across five random seeds.

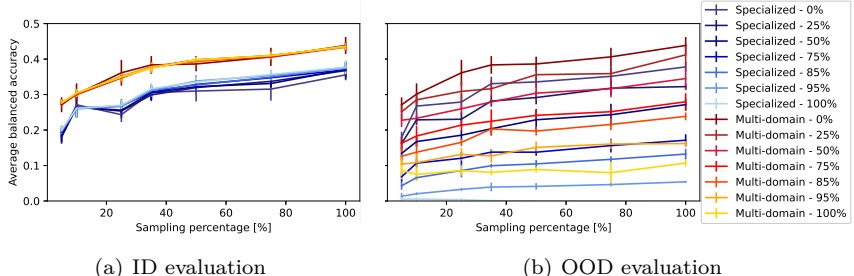

(a) ID evaluation          (b) OOD evaluation

Figure C.1: Comparison of models across various amount of data for ImageCLEFmedical. We report average balanced accuracy for specialized and multi-domain models across various sampling rates, depicted in x-axis, and OOD levels, indicated by different color codes, for both ID (a) and OOD (b) evaluation. We report the mean and standard deviation of the test accuracy across five random seeds.

