# OpenReview forum: "Multi-domain improves out-of-distribution and data-limited scenarios for medical image analysis"
_TMLR — Rejected by TMLR_

### Review · Reviewer_WPK7 · 2023-12-09

**Summary Of Contributions:**

Authors claim that combining data from different modes in a common setup can improve classification accuracy for unseen (out-of-distribution) data. Authors use PolyMNIST, MedMNIST and ImageCLEFmedical datasets as example to show the claim.

**Audience:**

No

**Claims And Evidence:**

No

**Requested Changes:**

(1) In the introduction section, all paragraphs before 'Related Work' should be re-written by very explicitly, directly, clearly pointing out the _specific_ contribution of this work, rather than presenting in a general and ambiguous narrative. Simply, state concrete, specific content, not in a general and vague tone.

- Example: “In this work, we introduce the incorporation of diverse medical image data domains and evaluate the performance of specialized models in comparison to multi-domain models.” What are data domains used in the paper? In what metric is the performance characterized? What are the baseline multi-domain models with which experiments are benchmarked?

- Example: “we focus on learning from different domains or datasets without necessariy involving multiple tasks”. What is learned? What is the concrete application? What is the performance? What is the advantage/gain over baselines?

Similarly, for related work, it is completly unnecessary to cite many prior work on NLP, CV and LLM without clearly specifying its concrete connection to the content of this work.

(2) Methodology:
 - There is actually no new methodology proposed in this manuscript at all.
 - Authors must clarify their actual intention of using words “modality” and “domain”. In 2.1.1, number digit images with different backgrounds are not from different “modalities”.

(3) Experiments:
 - Intuitively, for OOD samples, the classification results from a trained classifier are most likely to be a random guess. The so-called-claimed “multi-domain” training data setup is simply to include data of addtional modes into the training dataset, then the superior performance over models trained with “specialised data” is expected and trivial. The phrase “shared information” in the main text should be clearly explained.
 - Authors fail to specify how out-of-distribution (OOD) samples and in-distribution (ID) samples are partitioned/created with respect to each dataset. There must be crystal clear quantitative specification on the data setup.
Authors should make a table to lay out what categories/classes are used as IN and OOD respectively, and the data partition scheme for each setting (specialised, specialised upsampled, foundational).
 - It is unclear why the shown modes (a,b,c,d,e) in PolyMNIST are inherently different. There needs to be numerical evidence to show that data from different modes cannot be handled by some model trained with data only from some certain modes.
 - Authors should give very specific and clear expalantion about what input and output are respectively for the adopted model. If the output is not a logit, then area under the curve (AUC) metric needs further explanation about how it fits into the setting.
 - Authors should clearly explain what “balanced accuracy” means.
 - What does “sampling percentage” refer to? How are train-test split designed?
 - Define $\mu_{\textrm{digit}}$, $\sigma_{\textrm{digit}}$, $\mu_{\textrm{modality}}$, $\sigma_{\textrm{modality}}$.
 - In section 3.2.2 MedMNIST, authors claim “data distribution for MedMNIST shows a significant bias towards the axial view, particularly in the liver/axial organ/view combination”. Authors need to provide more specification to support this claim.
 - Intuitively, given that the data could be from several modes that we can pre-determine, then one can simple prepare corresponding copies of classifier to handle each mode. Authors should benchmark the performance of the so-claimed “multi-domain model” with mode-specific models in their respective domain.
 - In figures 4 and 5, the curves seem to start from the point with “number of samples=0”. Explain this issue.

Minor comments:

 (1) explain the meaning of “concept” in section 2.1.3.

 (2) what is “CUI” in section 2.1.3?

**Strengths And Weaknesses:**

Weakness:

- There is essentially no new methodology proposed in this manuscript.

- Ambiguity in statement is common regarding main technical details in this manuscript. Specific questions are specified in the Question section.

---

> ### Author Response · Authors · 2023-12-13
>
> Thank you for your comments on our manuscript. We would like to urgently clarify the concerns about the contributions and methodology.
>
> We introduce a new technique to mitigate OOD and limited data problems that consists of combining diverse medical image domains. We show that our approach substantially surpasses the conventional specialized training method used in healthcare. Our approach is simple but it is not trivial, as to the best of our knowledge we are the first to introduce it and show that it achieves significant accuracy improvements. Simplicity is a good thing following Occam razor.
>
> We appreciate the comments as they help us improve the paper and they will be addressed in a new version of the paper that we will upload in the next few days.

---

> ### Author Response · Authors · 2023-12-15
> **Response to the Reviewer WPK7 - part 1**
>
> Thank you very much for the valuable comments on our manuscript. We have carefully considered your feedback, and we hope that the revisions made have improved the manuscript. Below, we provide our point-by-point responses after each comment.
>
> > (1) In the introduction section, all paragraphs before 'Related Work' should be re-written by very explicitly, directly, clearly pointing out the specific contribution of this work, rather than presenting in a general and ambiguous narrative. Simply, state concrete, specific content, not in a general and vague tone.
>
> Within healthcare, clinicians often leverage various image domains, such as imaging modalities or viewpoints, to address different medical conditions or problems. However, current machine learning methods for medical image analysis primarily focus on developing models tailored for their specific tasks, utilizing data within their target domain. These specialized models tend to be data-hungry and often exhibit limitations in generalizing to out-of-distribution samples. In this work, we show that employing models that incorporate multiple domains instead of specialized ones significantly alleviates the limitations observed in specialized models. In order to better convey this better within our manuscript, we have revised Section 1 on p.1-2, removed the ambiguous parts and rephrased the text to have more explicit and direct information about the contributions.
>
> > Example: “In this work, we introduce the incorporation of diverse medical image data domains and evaluate the performance of specialized models in comparison to multi-domain models.” What are data domains used in the paper? In what metric is the performance characterized? What are the baseline multi-domain models with which experiments are benchmarked?
>
> We rewrote the second paragraph on p.1 to summarize the contributions of our work. More specifically, we added the different domains and the datasets we used. Furthermore, we added that the specialized models serve as the baseline to the multi-domain models.
>
> > Example: “we focus on learning from different domains or datasets without necessariy involving multiple tasks”. What is learned? What is the concrete application? What is the performance? What is the advantage/gain over baselines?
>
> To better explain the difference of multi-domain to multi-task learning, we rephrased the second paragraph of Section 1.1 on p.2. Furthermore, we added a practical example.
>
> > Similarly, for related work, it is completly unnecessary to cite many prior work on NLP, CV and LLM without clearly specifying its concrete connection to the content of this work.
>
> We rephrased the first paragraph of the Related Work section, where we removed the examples of LLM and CV foundation models for general applications and concentrated only on healthcare references. Furthermore, for vision foundation models we gave a more detailed overview. See on p.2 Section 1.1 first paragraph.
>
> > Authors must clarify their actual intention of using words “modality” and “domain”. In 2.1.1, number digit images with different backgrounds are not from different “modalities”.
>
> In this manuscript, the term "domain" denotes various data sources, which includes a range of diverse medical imaging modalities or viewpoints. Additionally, PolyMNIST dataset [Sutter et al., 2021] is a multi-modal benchmark and has been widely used as a toy dataset. We adopt the same terminology used by authors, where they refer to each background as a modality to capture source specific information. We added this note to the revised manuscript in Section 2.1.1 on p.3.

---

> ### Author Response · Authors · 2023-12-15
> **Response to the Reviewer WPK7 - part 2**
>
> > Intuitively, for OOD samples, the classification results from a trained classifier are most likely to be a random guess. The so-called-claimed “multi-domain” training data setup is simply to include data of addtional modes into the training dataset, then the superior performance over models trained with “specialised data” is expected and trivial. The phrase “shared information” in the main text should be clearly explained.
>
> We politely disagree with the reviewer that the behavior of multi-domain data is expected and trivial. Consider a medical scenario: would observing an organ through MR, CT, and US images help in classifying the same organ within an X-ray image, even when limited samples of that specific organ are available from X-ray in the dataset? This is essentially what the experiments on OOD levels for the multi-domain model evaluate. Our experimental results suggest that indeed, this multi-domain approach helps. Furthermore, for OOD samples the accuracy is indeed worse than random guess. From our perspective, this behavior isn't obvious or expected, as the “shared information” about the organ across diverse modalities isn't straightforward; in fact, it's rather surprising. By "shared information," we refer to the task-specific information that is commonly present across different modalities. For PolyMNIST, it is the digit specific information, for medical datasets it is the organ specific information across image views or modalities. In the revised manuscript on p. 9 (for PolyMNIST) and p.11 (for MedMNIST) we added this note about shared information.
>
> > Authors fail to specify how out-of-distribution (OOD) samples and in-distribution (ID) samples are partitioned/created with respect to each dataset. There must be crystal clear quantitative specification on the data setup. Authors should make a table to lay out what categories/classes are used as IN and OOD respectively, and the data partition scheme for each setting (specialised, specialised upsampled, foundational).
>
> We updated Figure 1 on p.4 to focus only on domain/task information. We updated the example in Section 2.4 for PolyMNIST on p. 6. Additionally, we included a new figure (Figure 4 in the revised manuscript) on p.7, showing training and evaluation methodologies for specialized and multi-domain models for PolyMNIST. The first three subfigures illustrate the specialized setup, while the latter three subfigures illustrate the multi-domain setup, featuring a 100% OOD level for digit 2 and modality d, ensuring that the models have no exposure to cases involving digit 2 and modality d. For the OOD evaluation, (ii, v), both models are evaluated on test samples featuring digit 2 from modality d. For the ID evaluation, (iii, vi), the specialized model is assessed using test samples from modality d and all digits except 2, while the multi-domain model undergoes evaluation for all other digit/modality combinations except digit 2 and modality d. We added this explanation on p.8, Section 3.1.2.
>
> > It is unclear why the shown modes (a,b,c,d,e) in PolyMNIST are inherently different. There needs to be numerical evidence to show that data from different modes cannot be handled by some model trained with data only from some certain modes.
>
> Our experiments involve training specialized models exclusively with data from their respective modalities. For instance, a specialized model for modality d is exclusively trained using data from modality d and evaluated on modality d, as illustrated in Figure 4(i-iii). As the reviewer suggested, we also evaluated the specialized model trained with data from modality d on the other modalities a,b,c,e. For this, we included different OOD levels for digit 2 and modality d. In summary, training on one domain and testing on another leads to large drops of accuracy. More specifically, for OOD level of 50%, the average balanced accuracy evaluated on digit 2 and domain d yielded 0.95, whereas on domains a,b,c,e yielded 0.71. Average balanced accuracy tested on all digits except digit 2 and on modality d yielded 0.94, in contrast, it was 0.77 for the modalities a,b,c,e. For OOD level of 85%, the average balanced accuracy evaluated on digit 2 and domain d yielded 0.87, whereas on domain a,b,c,e it was 0.59. Average balanced accuracy tested on all digits except digit 2 and on modality d was 0.94, in contrast it was 0.73 on domains a,b,c,e. We added this experiment and numbers to the manuscript on p. 10.

---

> ### Author Response · Authors · 2023-12-15
> **Response to the Reviewer WPK7 - part 3**
>
> > Authors should give very specific and clear expalantion about what input and output are respectively for the adopted model. If the output is not a logit, then area under the curve (AUC) metric needs further explanation about how it fits into the setting.
>
> Input: Different data distributions including OOD levels.
>
> Output: Each of the models is trained as a multi-class problem and the class with the highest probability is assigned as the estimated class.
>
> Evaluation: First, we calculate the OOD and ID average balanced accuracy for different OOD levels and amount of data in Figure 7, each curve corresponding to different OOD levels. To make the results more compact, we calculate the AUC, corresponding to the areas under the curves for 7 different OOD levels illustrated in Figure 7 and present this on Figure 5. Note that AUC is not the common AUC used in some object detection tasks.
>
> This is added to the revised manuscript on p. 9, Section 3.2.1.
>
> > Authors should clearly explain what “balanced accuracy” means.
>
> Average balanced accuracy corresponds to the mean accuracy across all task/domain combinations, which is different from overall accuracy or balanced accuracy. Our evaluation approach involves a detailed breakdown based on task and domain. For instance, in PolyMNIST, we calculate the accuracy for each digit/modality combination, resulting in 50 individual values (10 digits and 5 modalities). Subsequently, we compute the average of these values, which we refer to as the average balanced accuracy. This is similarly applied in OOD experiments. We incorporated this to the revised manuscript on p.8 Section 3.1.2.
>
> > What does “sampling percentage” refer to? How are train-test split designed?
>
> Number of images for each organ/view combination in MedMNIST for the respective train/validation/test splits are shown in Figure 2(a), wherein (i) shows the number of images in the train set (with 61521 images in total), (ii) validation set (with 11335 images in total), and (iii) test set (with 34875 images in total). Note that, for MedMNIST, we used the official train,validation and test splits. Furthermore, Figure 2(b) shows the number of images for each organ/modality combination in train+validation and test splits for ImageCLEFmedical, which was used in our work. For training+validation split, the number of samples are shown in (i). We divided the original challenge dataset's train split into training and validation sets with a 75/25% ratio. The validation split was repurposed as the test set, shown in (ii). These figures were initially placed later in the manuscript, leading to confusion. Thus, we have reorganized and relocated them to p. 4 for improved clarity.
>
> We conducted sampling percentage experiments on both MedMNIST and ImageCLEFmedical datasets. To illustrate, when employing a 50% sampling rate for MedMNIST, the models were exposed to 50% of the available data for each organ/view combination for training and validation. However, during testing, the entire test set was utilized for evaluation. Thus, the models utilized 50% of the total available training samples, amounting to 61521 * 0.5 = 30760 samples, 11335 * 0.5 = 5668 validation samples, and retained the full 34875 test samples for evaluation purposes. We added this example on p. 5/6, Section 2.3.1 in the revised manuscript for better clarity.
>
> > Define \mu_digit, \sigma_digit, \mu_modality, \sigma_modality.
>
> The parameters of the digit and modality distributions \mu_digit, \sigma_digit, \mu_modality, \sigma_modality represent the mean and standard deviations within standard distributions for the digit and modality distributions. These details have been incorporated into p.5 Section 2.3.1 of the revised manuscript.
>
> > In section 3.2.2 MedMNIST, authors claim “data distribution for MedMNIST shows a significant bias towards the axial view, particularly in the liver/axial organ/view combination”. Authors need to provide more specification to support this claim.
>
> Figure 2(a) displays the image distribution across organ/view combinations in MedMNIST's train/validation/test splits. Across all splits, the axial view consistently contains the highest number of images, notably with the liver in the axial view having the most images in both the train and test splits. We rephrased this sentence on p.12 Section 3.2.2 for better clarity.

---

> ### Author Response · Authors · 2023-12-15
> **Response to the Reviewer WPK7 - part 4**
>
> > Intuitively, given that the data could be from several modes that we can pre-determine, then one can simple prepare corresponding copies of classifier to handle each mode. Authors should benchmark the performance of the so-claimed “multi-domain model” with mode-specific models in their respective domain.
>
> The “mode-specific benchmark” is precisely the specialized approach in our paper. We clarify this in the Introduction (in the contributions), Section 2.4 and illustrated in Figure 4.
>
> > In figures 4 and 5, the curves seem to start from the point with “number of samples=0”. Explain this issue.
>
> Figure A.1 on p.18 displays the different image distributions corresponding to the number of samples along the x-axis in Figures 6 and 7 (in the revised manuscript). The x-axis data points are labeled as the titles of each subfigure, ranging from a minimum of 6 (representing the median of the distribution in the first row, first column) to a maximum of 890 (reflecting the median of the distribution in the second row, last column). We added this to the revised manuscript on p. 8, Section 3.2.1.
>
> > Minor comments:
>
> > (1) explain the meaning of “concept” in section 2.1.3.
>
> Concepts are from Unified Medical Language System (UMLS) and they define different terms. For instance, "plain x-ray" is a concept identified by the CUI C1306645, while "pelvis" is another concept denoted by the CUI C0030797. We added this example to the revised manuscript on p.4, Section 2.1.3.
>
> > (2) what is “CUI” in section 2.1.3?
>
> CUI refers to Concept Unique Identifiers from Unified Medical Language System (UMLS) concepts. We added this description to the revised manuscript on p.4, Section 2.1.3.

---

### Review · Reviewer_vTYw · 2023-12-19

**Summary Of Contributions:**

This paper evaluates “multi-domain” training, referring to training for the same task across different image modalities and views, in the setting of medical image tasks. The evaluation was conducted on PolyMNIST where multiple domains come from different background images on digit images, MedMNIST data where different views of CT images are considered, and ImageCLEFmedical where multiple medical image modalities are considered. Experiments considered both in-distribution and out-of-distribution performances at different levels of data sparsity and distribution.

**Audience:**

Yes

**Broader Impact Concerns:**

No concerns

**Claims And Evidence:**

Yes

**Requested Changes:**

- Add investigations into multi-domain single task models that explicitly addresses domain differences.

- Add analysis and interpretation of the representations being learned across domains.

- Increase the difference of data distribution (especially in terms of imbalance across content/domain combinations) and the extent to which OOD is applied to the percentage of different content/domain combinations.

- Descriptions in 2.3.1 about data diversity and distribution was difficult to understand, especially the reference to the mean and sigma of digit and modality distributions in PolyMNIST (and similarity in 2.3.2). Fig (2)a and Fig A1 are difficult to understand — what represents training vs. validation differences?

- The training, validation, and test distribution as illustrated in Fig 2 c-d appear similarly overall?

- Fig 4 seems be a summary of Fig 5 (left column - right column); if that’s the case, may not be necessary to include both figures (or at least move one of them to appendix)

- It seems that the advantage of multi-domain foundation model only shows up In OOD tasks when OOD level exceeds 80%. Please comment on the anticipated benefit of the multi-domain model when there is no such extreme OOD, especially considering that the training of such a model requires more resources compared to a specialized model.

-The OOD setting seems to be in generally applied on a small subset of the instances (as shown in Fig 2). Please comment and experiment on the extent to which OOD can be applied to different combinations in the data.

**Strengths And Weaknesses:**

Strengths:

- The topic presented was not investigated systematically before and thus the findings are of value to the community.

- The experimental setups and results are relatively comprehensive, considering different datasets, different combinations of domains, and a focus on different levels of OOD difficulties.


Weaknesses

- The overall insights derived seem to be limited — the multi-domain model, when seeing a much larger number of data across different combination of domains, seems to be expected to be more generalizable compared to a specialized model. How does this type of multi-domain model compare to models trained across the same combinations of data, but including additional strategies (such as domain invariance etc) to better integrate multi-domain information? Results along this line would be more interesting and need to be added.

- More interpretations and insights about the representations learned across such domains could be added.

- The train/validation/test distribution of data (in terms of imbalance) seems to be similar and their difference could be more substantial. The amount of OOD level, in terms of the percentage of combinations being impacted, could be increased too.

---

> ### Author Response · Authors · 2024-02-07
> **Response to the Reviewer vTYw - part 1**
>
> Thank you very much for the valuable comments on our manuscript. We have carefully considered your feedback which helped a lot to improve the manuscript. Below, we provide our point-by-point responses to the comments.
>
> > Add investigations into multi-domain single task models that explicitly addresses domain differences.
>
> This paper aims to introduce a multi-domain strategy and compare its performance with the conventional single domain specialized approach, particularly within the context of medical image analysis, where the latter is the norm. The advantage of the multi-domain model lies not only in its access to a larger amount of data but also in its ability to leverage a diverse range of data domains. To illustrate this difference, our experiments evaluated a modified approach, termed specialized upsampled, using the PolyMNIST toy dataset to evaluate a higher amount of data with the specialized approach. Our results show that the strength of the multi-domain model does not only lie at the higher amount of data but the robust representations that the model learns. We acknowledge the reviewer's suggestion regarding the potential for improving multi-domain models by incorporating domain-specific knowledge into the training process, e.g. through the use of alternative loss functions. However, this line of work falls outside the scope of our current work and would be a very interesting future direction. We have included this discussion on p. 14 for further exploration.
>
> > Add analysis and interpretation of the representations being learned across domains.
>
> We conducted the experiment suggested by the reviewer to analyse the representations being learned across domains and to assess the robustness of the learned representations to domain changes. Specifically, we examined the performance of a specialized model trained on data from modality d when applied to other modalities a, b, c, e to evaluate the robustness of the learned features. To do this, we varied the OOD levels for digit 2 and modality d. In summary, when training on one domain and testing on another, we observed significant drops in accuracy. For instance, at an OOD level of 50%, the average balanced accuracy for digit 2 and domain d was 0.95, compared to 0.71 for domains a,b,c,e. Similarly, when testing on all digits except digit 2 and on modality d, the average balanced accuracy was 0.94, whereas it was 0.77 for modalities a,b,c,e. At an OOD level of 85%, the average balanced accuracy for digit 2 and domain d was 0.87, while it dropped to 0.59 for domains a,b,c,e. Likewise, the average balanced accuracy for testing on all digits except digit 2 and on modality d was 0.94, contrasting with 0.73 for domains a,b,c,e. In contrast, the multi-domain model learns a single model across all domains, is more robust to different domains, which results in OOD generalization, as shown in Fig. 6(b) and 7(f). These findings, along with the corresponding numerical results, have been incorporated into the manuscript on page 10.
>
> > Increase the difference of data distribution (especially in terms of imbalance across content/domain combinations) and the extent to which OOD is applied to the percentage of different content/domain combinations.
>
> In our experiments, PolyMNIST data was sampled across 24 diverse data distributions, ranging from highly sparse (as shown in the first row of Fig. A1) to flatter distributions (as shown in the second row of Fig. A1) for digit/modality combinations. The evaluation of each of these distributions is presented in Figures 5-7 in the revised manuscript. To specifically assess the impact of a flat distribution on the number of samples, an additional experiment is conducted and illustrated in Figure A2. For MedMNIST, given its inherent imbalanced distribution, we initially assessed the available data by sampling while maintaining the data distribution, as reflected in Figures 9-10. Subsequently, another experiment was conducted using a flat distribution for organ/view combinations, and the results are presented in Figure 11. In the case of ImageCLEFmedical, the assessment of data availability involved sampling the existing data while preserving the data distribution, as demonstrated in Figures 12-13. Thus, our experiments comprehensively evaluated data availability from both the quantity of data and the data distribution perspective.
>
> Regarding out-of-distribution (OOD) evaluations, we systematically explored a broad spectrum of OOD levels, ranging from 0% (no OOD) to 100% (full OOD). To isolate and facilitate a more focused comparison of model strengths or drawbacks, we intentionally conducted OOD evaluations for a single task/modality combination, opting for a focused approach, rather than having OOD evaluations for multiple combinations.

---

> ### Author Response · Authors · 2024-02-07
> **Response to the Reviewer vTYw - part 2**
>
> > Descriptions in 2.3.1 about data diversity and distribution was difficult to understand, especially the reference to the mean and sigma of digit and modality distributions in PolyMNIST (and similarity in 2.3.2). Fig (2)a and Fig A1 are difficult to understand — what represents training vs. validation differences?
>
> We've refined the descriptions of digit and modality distributions in Section 2.3.1 on p. 5 of the revised manuscript for better clarity. Specifically, we provided detailed explanations of normal distribution parameters for both digits and modalities. Additionally, to better explain Section 2.3.2, we included Figure 3(d) on p.6. Regarding the training and validation splits, Figures 3(c) and A.1 display distributions utilized for training+validation, where the presented data distribution is split into training and validation sets with a 0.75 ratio; thus, the train and validation sets have the same data characteristic. For the test set, however, a uniform distribution is evaluated, where each digit/modality combination has the same amount of samples, as explained in Section 2.1.1.
>
> > The training, validation, and test distribution as illustrated in Fig 2 c-d appear similarly overall?
>
> For the medical experiments, MedMNIST and ImageCLEFmedical, we utilized the official splits provided. For the amount of data experiments, we sampled the training and validation data, maintaining consistent ratios of training and validation samples across different combinations, albeit with varying sample sizes. For testing, the distribution or the amount of data are fixed for all experiments. We incorporated an example for MedMNIST on p. 5-6 in the revised manuscript to increase the clarity. The experimental setup for the ImageCLEFmedical dataset follows a similar design.
>
> > Fig 4 seems be a summary of Fig 5 (left column - right column); if that’s the case, may not be necessary to include both figures (or at least move one of them to appendix)
>
> Indeed, it's the summary. In the revised manuscript, we relocated these figures, where both Figures 5 and 6 are a summary of Figure 7. To provide a clearer understanding of the summary, we displayed a detailed figure specifically for PolyMNIST, omitting the detailed figures for the medical datasets and relocating them to the appendix. However, in the interest of enhancing comprehension of the summary, we believe retaining the detailed figure for at least one dataset is crucial.
>
> > It seems that the advantage of multi-domain foundation model only shows up In OOD tasks when OOD level exceeds 80%. Please comment on the anticipated benefit of the multi-domain model when there is no such extreme OOD, especially considering that the training of such a model requires more resources compared to a specialized model.
>
> An OOD level exceeding 80% highlights situations where the model encounters class/domain combinations either extremely rarely or never before. Such scenarios are common in medical applications, e.g. involving rare diseases or conditions. Accessing medical conditions from diverse data domains could be beneficial, especially if the condition hasn't been observed frequently within a specific domain in the training data. Considering the uncertainty about the OOD level in a real-case scenario, the multi-domain model navigates a trade-off for the necessary resources. Furthermore, in data-limited scenarios evaluated in our experiments, employing a multi-domain model presents greater benefits without imposing significant data bottlenecks during training for these restricted data scenarios. We've included this discussion in the revised manuscript on p.14.
>
> > The OOD setting seems to be in generally applied on a small subset of the instances (as shown in Fig 2). Please comment and experiment on the extent to which OOD can be applied to different combinations in the data.
>
> We acknowledge the reviewer's observation that the OOD setting was implemented on a limited subset of instances. Specifically, for each experiment, we deliberately applied the OOD setting only once. This intentional choice was made to ensure a fair representation  of each task/modality combination. To provide more clarity, each data point in Figure 7 represents the average accuracy obtained from 50 distinct experiments. In these experiments, one task (out of 10 tasks: digits 0 to 9) and one modality (out of 5 modalities: a to e) were systematically excluded from training, and this process was repeated 50 times to encompass all possible task/modality combinations. We added this explanation to the revised manuscript on p.9.

---

### Review · Reviewer_MEyy · 2024-04-29

**Summary Of Contributions:**

Deep neural networks for medical image classification usually require large datasets from the target domain, and cannot always generalize to OOD samples. To overcome these challenges, the authors propose a multi-domain model that uses images from multiple domains, compared to a single one. They train a ResNet-18 with three publicly available datasets: PolyMNIST, MedMNIST, and ImageCLEFmedical, and refer to three different models based on the training datasets: specialized (supervised), specialized upsampled (supervised + augmentations), and multi-domain. They evaluate AUC and average accuracy across all scenarios.

**Audience:**

Yes

**Broader Impact Concerns:**

Not applicable.

**Claims And Evidence:**

No

**Requested Changes:**

Please see above under weaknesses, and the following:
- Some in-text citations need to be fixed, such as in the contributions section: (Sutter et al., 2021) instead of Sutter et al. (2021)
- Some sentences are unclear, for example: “Specifically, it demonstrates an example of utilized data to train for evaluating data distribution and employ distribution parameters μdigit = 0, σdigit = 5, μmodality = 2, and σmodality = 3 to define the data distribution”
- The paper needs to be proofread to improve grammar/ spelling and other presentation issues: such as works should be work, reproducability, use of quotations, etc.
- I am not sure if Figure 3d fits with the overall figure. It might be better to just reference Figure 4i to explain OOD levels.
- The use of a, b, c vs I, ii, and iii in the Figures - please ensure consistency
- How did they perform hyperparameter tuning? This must be clarified.
- Why use five random seeds for MedMNIST and ImageCLEFmedical only? What about PolyMNIST?

**Strengths And Weaknesses:**

**Strengths:**
- The paper illustrates the advantage of training with a diverse dataset (multiple modalities/ views) compared to a dataset with a single modality / view.
- They run experiments with three publicly available datasets.
- They perform multiple comparisons in each experimental setting.

**Weaknesses:**
- The study is primarily experimental and the presented findings are expected. Although novelty is not necessarily expected, it does come off as a technical report that summarizes results of several experiments.
- The related work section discusses foundation models, multi-task learning and multi-modal learning, but not multi-domain models, see https://arxiv.org/pdf/1803.10082 or https://ieeexplore.ieee.org/document/9356259. Hence the study needs to be contextualized with respect to studies that also aim to develop similar approaches under multi-domain learning. This can support the statement: "This work represents the first instance where multi-domain data is assessed for a single task, different from multi-task setups. "
- "We will make our code publicly available upon acceptance" - can you share an anonymized draft of the repository?
- Convert Figure 2 to a table - not intuitive in its current format, or provide a Table with the exact numbers in the appendix.
-There are several parts that need to be rewritten as it's hard to understand what is intended, such as: (i) Our experiments exploit the concept detection data within this dataset" What does this mean? How are the concepts used in the experiments? (ii) The datasets can be visualized as a square task and data domain combinations grid." What does this mean? Is this referring to the Figure?
- "in our work average balanced accuracy corresponds to the mean accuracy across all task/domain combinations, which is different from overall accuracy or balanced accuracy." — why then used the term 'balanced'? Isn't this just average accuracy?
- The authors refer to multi-domain as foundation model in one of the figures, why is this the case? The model should be labelled consistently and I don't think this qualifies as a foundation model.
- The AUC values across figures vary from 0-1000 and 0-100 -> it should be between 0 and 1.
- The error bars should be provided for the average accuracy across all figures.
- The authors should perform statistical significance testing to support statements like "especially significant".
- In Figure A1, using term "datapoint" to represent the median - clarify in the figure it is unclear
- How did the authors obtain the black dashed line in Figure 6? Which model/ setup? Similarly in Figure 9, is this just to indicate perfect AUC or was it obtained by a different model?
- There is some repetition across the paper, both in the text and figures. For example, Figure 6 shows the differences obtained from Figure 7.
- Also are those differences of multi-domain minus specialized, or vice versa?
- In Figure 7, there is a strange peak at a small number of samples for the multi-domain model, can you investigate this further? Shouldn’t it be increasing monotonically?

---

> ### Author Response · Authors · 2024-05-06
> **Response to the Reviewer MEyy - part 1**
>
> Thank you very much for the valuable comments on our manuscript. We have carefully considered your feedback which helped a lot to improve the manuscript. Below, we provide our point-by-point responses to the comments.
>
> > The study is primarily experimental and the presented findings are expected. Although novelty is not necessarily expected, it does come off as a technical report that summarizes results of several experiments.
>
> We politely disagree with the reviewer that the behavior of multi-domain data is expected. Consider a medical scenario: would observing an organ through MR, CT, and US images help in classifying the same organ within an X-ray image, even when limited samples of that specific organ are available from X-ray in the dataset? This is essentially what the experiments on OOD levels for the multi-domain model evaluate. Our experimental results suggest that indeed, this multi-domain approach helps. From our perspective, this behavior isn't obvious or expected, as the shared information about the organ across diverse modalities isn't straightforward; in fact, it's rather surprising.
>
> > The related work section discusses foundation models, multi-task learning and multi-modal learning, but not multi-domain models, see https://arxiv.org/pdf/1803.10082 or https://ieeexplore.ieee.org/document/9356259. Hence the study needs to be contextualized with respect to studies that also aim to develop similar approaches under multi-domain learning. This can support the statement: "This work represents the first instance where multi-domain data is assessed for a single task, different from multi-task setups. "
>
> We appreciate the reviewer for bringing this to our attention. In response, we have included a paragraph in the Related Work section discussing multi-domain models and highlighting the distinctions of our approach compared to recent works in the revised manuscript on p.3.
>
> > "We will make our code publicly available upon acceptance" - can you share an anonymized draft of the repository?
>
> We have made the draft of our code publicly available in this repository: https://anonymous.4open.science/r/multi_domain_medical-6686/. This repository includes notebooks for generating all the figures in the manuscript and example .sh scripts to execute all the experiments. The link to the repository has been added to the manuscript on p.3.
>
> > Convert Figure 2 to a table - not intuitive in its current format, or provide a Table with the exact numbers in the appendix.
>
> For precise numerical details corresponding to Figure 2(a), we included Table  B.1 in the appendix on p.20. For details pertaining to Figure 2(b), we added to Table C.1 in the appendix on p.24.
>
> > There are several parts that need to be rewritten as it's hard to understand what is intended, such as: (i) Our experiments exploit the concept detection data within this dataset" What does this mean? How are the concepts used in the experiments? (ii) The datasets can be visualized as a square task and data domain combinations grid." What does this mean? Is this referring to the Figure?
>
> (i) Note that, the ImageCLEFmedical Caption challenge comprises two subtasks: concept detection and concept prediction and our experiments utilize the concept detection data from this dataset. The concepts are then used to select the subset of data with organs and imaging modalities. We added this note in the revised manuscript on p.5.
>
> (ii) Each dataset comprises different tasks and data domains. This can be seen as a grid structure illustrated in Figure 1. Each square within this grid represents a unique task/domain combination. We rephrased this in the revised manuscript on p.5.
>
> > "in our work average balanced accuracy corresponds to the mean accuracy across all task/domain combinations, which is different from overall accuracy or balanced accuracy." — why then used the term 'balanced'? Isn't this just average accuracy?
>
> Accuracy is computed across all samples from all classes, while balanced accuracy averages the accuracy across all classes, considering both majority and minority classes irrespective of their data domain. In our study, the average balanced accuracy is calculated with a finer granularity: accuracy is calculated for each class within each data domain and then averaged. This approach ensures a fairer comparison across both majority and minority classes and their respective data domains.
>
> > The authors refer to multi-domain as foundation model in one of the figures, why is this the case? The model should be labelled consistently and I don't think this qualifies as a foundation model.
>
> Thank you for pointing that out. We have corrected the labeling of models to ensure consistency, referring to them consistently as either ‘specialized’ or ‘multi-domain’ models throughout the revised manuscript.

---

> ### Author Response · Authors · 2024-05-06
> **Response to the Reviewer MEyy - part 2**
>
> > The AUC values across figures vary from 0-1000 and 0-100 -> it should be between 0 and 1.
>
> The AUC values shown in Figure 5 are computed as the area under the curves from Figure 7. The maximum average balanced accuracy can reach 1 and the total number of samples can be maximum 890 (based on Figure A.1), resulting in a maximum AUC of 890. Similarly, for Figures 9, 11, and 12, where the maximum balanced accuracy can be 1 and the maximum sampling accuracy is 100%, the AUC ranges between 0 and 100 in these figures.
>
> > The error bars should be provided for the average accuracy across all figures.
>
> Error bars are presented in Figures B.1, B.4, and C.1 to facilitate comparison between specialized and multi-domain models across MedMNIST and ImageCLEF datasets. We reported the mean and standard deviation of test accuracy across five random seeds. However, for plots depicting AUC and average balanced accuracy differences, we chose not to include error bars to enhance plot clarity and comprehension.
>
> > The authors should perform statistical significance testing to support statements like "especially significant".
>
> We changed the wordings for using the word significant:
>
> p.1: … enhancing the overall outcomes substantially.
>
> P.14: …of multi-domain model is particularly pronounced for OOD tasks…
>
> > In Figure A1, using term "datapoint" to represent the median - clarify in the figure it is unclear
>
> In Figures 6 and 7, each datapoint on the x-axis represents a different distribution illustrated in Figure A.1. To be able to summarize and visualize these, we calculated the median of each of the 24 distributions, resulting in 24 values that correspond to the x-axis values in Figures 6 and 7. We added this to the revised manuscript on p. 19.
>
> > How did the authors obtain the black dashed line in Figure 6? Which model/ setup? Similarly in Figure 9, is this just to indicate perfect AUC or was it obtained by a different model?
>
> In Figure 5, the dashed line represents the maximum achievable AUC by the models. This is calculated using the highest possible average balanced accuracy, which can reach 1, and the total number of samples, maximum of 890 (as shown in Figure A.1). Consequently, the dashed line reflects a maximum AUC of 890. Similarly, in Figures 9, 11, and 12, where the maximum balanced accuracy can reach 1 and the maximum sampling accuracy is 100%, the highest AUC that a model can achieve is 100 in these figures. These black dashed lines do not indicate the accuracy of a trained model but rather serve as a ceiling for comparison. We added this note to the revised manuscript on p.8-9.
>
> > There is some repetition across the paper, both in the text and figures. For example, Figure 6 shows the differences obtained from Figure 7.
>
> We acknowledge the reviewer's observation that Figures 5 and 6 are derived from Figure 7. The inclusion of these figures was intentional to demonstrate how to derive AUC (Figure 5) and average balanced accuracy difference (Figure 6) plots from the detailed average balanced accuracy plot (Figure 7). Please note that the detailed average balanced accuracy plots for MedMNIST and ImageCLEFmedical datasets are not included in the main manuscript and are only available in the appendix. Furthermore, we rephrased the paragraph on p.8-10 to avoid repetitions in text.
>
> > Also are those differences of multi-domain minus specialized, or vice versa?
>
> The differences are computed as multi-domain minus specialized, resulting in positive differences in most cases, indicating that specialized models can achieve higher accuracy. This information has been included in the revised manuscript on p.10.
>
> > In Figure 7, there is a strange peak at a small number of samples for the multi-domain model, can you investigate this further? Shouldn’t it be increasing monotonically?
>
> We suspect that this peak at the small number of samples is due to the double descent phenomena. This is because the training error and the loss function in our experiments is going down, however for small number of samples the test error increases. We added this note to the revised manuscript on p. 11.
>
> > Some in-text citations need to be fixed, such as in the contributions section: (Sutter et al., 2021) instead of Sutter et al. (2021)
>
> We have reviewed the manuscript and updated some of the in-text references accordingly.
>
> > Some sentences are unclear, for example: “Specifically, it demonstrates an example of utilized data to train for evaluating data distribution and employ distribution parameters μdigit = 0, σdigit = 5, μmodality = 2, and σmodality = 3 to define the data distribution”
>
> We have revised the final paragraph of Subsection 2.3.2 on p. 7 to enhance clarity. Additionally, we have thoroughly reviewed the manuscript to address the specific issue raised by the reviewer.

---

> ### Author Response · Authors · 2024-05-06
> **Response to the Reviewer MEyy - part 3**
>
> > The paper needs to be proofread to improve grammar/ spelling and other presentation issues: such as works should be work, reproducability, use of quotations, etc.
>
> We have thoroughly reviewed the manuscript to address the presentation issues raised by the reviewer.
>
> > I am not sure if Figure 3d fits with the overall figure. It might be better to just reference Figure 4i to explain OOD levels.
>
> The image showing the data distribution in Figure 3(d) and Figures 4(i) and 4(iv) are identical, and this choice was intentional. We used Figure 3(d) initially to explain OOD. Subsequently, we used Figures 4(a) and 4(d) in the revised manuscript to illustrate the differences between specialized and multi-domain models. We prefer to maintain this approach to facilitate understanding of the sequential steps for the reader.
>
> > The use of a, b, c vs I, ii, and iii in the Figures - please ensure consistency
>
> We have updated the figure captions such that the first level is labeled as (a), (b), etc., and the second level is labeled as (i), (ii), etc.
>
> > How did they perform hyperparameter tuning? This must be clarified.
>
> For hyperparameter tuning, we employed multi-domain models and conducted a grid search for optimizing learning rate, weight decay and different ResNet architectures such as Resnet18, Resnet34, Resnet50. We then repeated this process for specialized models, with the hyperparameters largely aligned, ensuring that they did not affect the training convergence. We added this note on p.8 of the revised manuscript.
>
> > Why use five random seeds for MedMNIST and ImageCLEFmedical only? What about PolyMNIST?
>
> In our PolyMNIST experiments, we reported results from a single seed because our evaluations demonstrated that the dataset is highly stable and the results do not exhibit notable variation across different seeds. Similarly, results from MedMNIST also indicate minimal variability across different seeds, mirroring the stability observed in PolyMNIST.

---

### Decision · Action_Editor_wZan · 2024-06-10

**Recommendation:** Reject

**Comment:**

I can't see where realistically the scientific contribution of this paper is.

**Audience:**

Probably not. The overall insights in this paper seem to be limited.

**Claims And Evidence:**

I apologize for taking so much time to finalize the decision. The reviews for this paper were somewhat unusual, so I had to read this paper carefully myself to build my own view.

I believe the experiments are generally executed correctly. However, I fail to see the contribution of this paper to the body of knowledge on ML clearly enough that it should be accepted. While I can't point to a specific paper that completed equivalent experiments, I don't think this is necessarily because this idea is so novel that no such paper exists. I believe this is because different ad hoc methods for combining data sets are so frequently used by machine learning practitioners that they are considered "machine learning folklore" rather than a direction for research. In case I severely misunderstood this paper, my advice is to rewrite it such that it is clearer where the scientific insight of the proposed method is.